# Subglacial hydrology regulates oscillations in marine ice streams

Marianne Haseloff<sup>1</sup>, Ian J. Hewitt<sup>2</sup>, and Richard F. Katz<sup>3</sup>

<sup>1</sup>Department of Geoscience, University of Wisconsin-Madison, Madison, WI 53706, USA

<sup>2</sup>Mathematical Institute, University of Oxford, Oxford OX2 6GG, UK

<sup>3</sup>Department of Earth Sciences, University of Oxford, Oxford OX1 3QR, UK

**Correspondence:** Marianne Haseloff (mhaseloff@wisc.edu)

Abstract. Marine ice stream dynamics are sensitive to conditions at the grounding line and basal shear stress. Variations in subglacial hydrology have been implicated in ice stream speed-up and shutdown. To investigate the interplay between marine ice stream flow and subglacial hydrology, we couple models of a marine ice stream and a subglacial drainage system. The coupled system evolves dynamically due to a positive feedback between ice flow, heat dissipation at the ice stream bed, and basal lubrication. Our results show that depending on the hydraulic conductivity of the bed, distinct dynamic regimes can be identified. These regimes include steady streaming, hydraulically controlled oscillations, and thermally controlled oscillations. Periodic fast flow can be initiated by activation waves travelling upstream or downstream or quasi-simultaneously everywhere. Different dynamical regimes are characterised by large differences in grounded ice volume, even under modest changes of grounding line positions. These results imply a strong dependence of marine ice-sheet dynamics on evolving hydrological conditions at the bed and highlight the importance of a better understanding of subglacial hydrology.

#### 1 Introduction

Much of the West Antarctic Ice Sheet rests on a bed that is below sea level, making it a marine ice sheet. Most of its ice discharge occurs in regions of fast flow with velocities in excess of 100 m yr<sup>-1</sup> (called ice streams) embedded in regions of slow flow with velocities of only a few meters per year (Rignot et al., 2011). While most of West Antarctica has experienced dramatic mass loss in the last four decades, the Siple Coast region has a net positive mass balance (Rignot et al., 2019, 2022). This is due to the current inactivity of Kamb ice stream, which slowed down approximately 130 years ago (Retzlaff and Bentley, 1993) and the ongoing slow down of Whillans, Mercer, and MacAyeal ice streams (Stephenson and Bindschadler, 1988; Bindschadler and Vornberger, 1998; Joughin et al., 2002; Rignot et al., 2022). On longer time-scales, observations suggest that marine ice stream flow speeds vary on time scales of hundreds to thousands of years and can undergo shut-down-and-reactivation cycles (Hulbe and Fahnestock, 2007; Catania et al., 2012). Our aim in this paper is to better understand what role subglacial hydrology plays in these processes.

Some studies have suggested that subglacial hydrology is responsible for the speed up and slow down of Antarctic ice streams (Anandakrishnan and Alley, 1997; Elsworth and Suckale, 2016). Observations support the existence of a dynamic subglacial drainage system beneath West Antarctica's ice streams (Engelhardt and Kamb, 1997; Fricker et al., 2007). However, existing

observational work does not address the interplay between subglacial hydrology and marine ice sheet dynamics because of a lack of direct observation.

Conversely, theoretical studies of marine ice sheet dynamics usually focus on static basal boundary conditions (e.g., Weertman, 1974; Schoof, 2007a, b, 2012; Tsai et al., 2015; Joughin et al., 2019; Sergienko and Wingham, 2019; Sergienko and Haseloff, 2023; Gregov et al., 2023) or prescribe their evolution (Brondex et al., 2017; Sergienko and Wingham, 2024). These studies show that the dynamics of the grounding line, which delineates the transition between grounded and floating ice, are sensitive to the basal boundary conditions. However, without an evolving subglacial drainage system, these models do not investigate which dynamic feedbacks exist between marine ice-stream dynamics and subglacial drainage.

Investigation of the interaction between ice-stream flow and subglacial drainage raises the question of how regions of fast and slow flow can co-exist in close proximity. One answer to this question invokes a positive feedback between heat dissipation and ice flow. Different mechanisms for this feedback have been hypothesised, relating it to either viscous weakening or to basal heat dissipation (e.g., Clarke et al., 1977; Fowler and Johnson, 1996; Payne and Dongelmans, 1997; Bougamont et al., 2011; Robel et al., 2014; Kyrke-Smith et al., 2014, 2015; Brinkerhoff and Johnson, 2015; Schoof and Mantelli, 2021). In this study, we focus on the latter feedback.

If the basal temperature is at the melting point, rapid sliding is possible and heat dissipation can be focused at the bed. The dominant positive feedback mechanism then involves melting at the base of the ice; faster sliding leads to more heat dissipation, which in turn produces additional melt water that reduces basal drag and permits even faster sliding until other processes suppress further weakening of the bed. For this positive feedback to operate, melt water production must raise the water pressure at the bed, and hence reduce the effective pressure, defined as the difference between ice overburden and basal water pressure. This behaviour can be caused by two mechanisms. In the first, the effective pressure decreases through an increase in water storage at the bed. If water storage is in subglacial till and no drainage is possible due to the material's low permeability, this is called the undrained bed model (e.g., Tulaczyk et al., 2000b; Sayag and Tziperman, 2008). In the second mechanism, an increase in subglacial water discharge reduces the effective pressure (as is common for distributed hydrologic systems, e.g., Fowler and Johnson, 1996; Kyrke-Smith et al., 2014).

Several studies using the undrained bed model have shown that the positive feedback between fast flow and lubrication can explain oscillatory behaviour of ice streams when modulated by a thermodynamic feedback (MacAyeal, 1993; Fowler and Johnson, 1996; Payne and Dongelmans, 1997; Fowler and Schiavi, 1998; Bougamont et al., 2003; Christoffersen and Tulaczyk, 2003; Robel et al., 2013, 2014; Feldmann and Levermann, 2017). Faster ice flow results in dynamic thinning. When the ice thickness becomes too small, the amount of conductive cooling at the bed exceeds the heat dissipation due to flow, water freezes, and the ice stream shuts down. However, these studies neglect the transport of subglacial water and its effect on water content of the till.

Fewer theoretical studies investigate the role of the second process (i.e., of subglacial hydrology), and these studies do not account for the subglacial conditions at the grounding line (Fowler and Johnson, 1996; Kyrke-Smith et al., 2014). Nevertheless, large-scale models capable of including these dynamics are beginning to emerge (e.g., Kazmierczak et al., 2022, 2024), motivating a better understanding of these processes. An extreme example of a subglacial drainage model assumes that subglacial

hydrology is so efficient that hydraulic gradients cannot develop. Steady grounding lines are possible in this case (Tsai et al., 2015).

The goal of this study is to investigate the simplest possible feedback between fast flow, heat dissipation, and basal lubrication, and its role in marine ice sheet dynamics. We use a model that includes the positive feedback between fast flow and basal lubrication through both storage of water in the subglacial sediment as well as evacuation of water through an active subglacial drainage system. Subglacial water flow is modelled with a Darcy-style flux law. The hydraulic conductivity is the quotient of a conductivity factor and the effective pressure, i.e., it is assumed to increase as the effective pressure decreases, as is common for distributed systems. The overall balance of storage and subglacial discharge is determined by the hydraulic conductivity factor. At very low hydraulic conductivity ( $K \ll 10^{-3} \text{ m s}^{-1}$ ), the model reproduces the undrained bed model, and water content of the bed is determined by the local energy balance. At very high hydraulic conductivities ( $K \gg 10^{-3} \text{ m s}^{-1}$ ), hydraulic gradients cannot develop at the bed; in this limit the effective pressure at the bed is set by the ice and bed geometry alone, independent of the local melt rate. This is the limiting case considered in Tsai et al. (2015).

The main novelty of our study is the investigation of marine ice sheet dynamics over the entire spectrum of different hydraulic conductivities of the subglacial system. This puts the results of existing studies with vanishing (Robel et al., 2014) and infinite (Tsai et al., 2015) hydraulic conductivities into context. We find that at intermediate hydraulic conductivities of the bed, the interplay between local water storage and subglacial drainage can lead to oscillations with distinct character and frequency. These hydraulically induced dynamics differ from those previously reported (MacAyeal, 1993; Robel et al., 2014); they are controlled by the efficiency of the subglacial drainage system, rather than the local melt rate at the bed.

# 2 The Model

We formulate the simplest possible model with an evolving hydraulic system and grounding line dynamics. Naturally, such a model must parameterize or neglect some other important aspects of ice-sheet dynamics and we discuss the limitations of our model below in section 4.4.

The computational domain is of length  $L_x=1000$  km with a downward sloping bed at elevation  $b=z_0-z_1x$ . All parameters are listed in table 1. We solve for velocity u, ice thickness H, grounding line position  $x_g$ , water content of the bed  $h_w$  (given as water column thickness), and thickness of unexpanded, hydraulically active till layer  $h_s$ , which can evolve due to freezing and melting in the sediment, see figure 1.

## 2.1 Ice mechanics

We use a standard, depth- and width-integrated flowline model for ice sheet dynamics (Dupont and Alley, 2005; Nick et al., 2009; Robel et al., 2014; Haseloff and Sergienko, 2018). The ice thickness H evolves according to the equation for mass balance,

90 
$$\frac{\partial H}{\partial t} + \frac{\partial (Hu)}{\partial x} = a,$$
 (1)

Figure 1. Model geometry and details of freezing. A marine ice sheet of thickness H and width W flows on a downwards sloping bed, becoming afloat at the grounding line. The underlying sediment is of thickness  $(e+1)h_s + (e_c+1)(h_0-h_s)$  with e the void ratio of the unfrozen sediment,  $e_c$  the void ratio at which the sediment freezes,  $h_0$  the thickness of the sediment layer if all void space was removed from it (i.e., at e=0), and  $(e+1)h_s$  the thickness of the expanded, hydraulically active, unfrozen portion of the sediment. At effective pressures N below the critical effective pressure  $N_c$ , freezing is focused at the ice-bed contact (panel a). At  $N \ge N_c$ , freezing can intrude into the void space between sediment grains, and a frozen fringe forms (panel b).

with u the ice velocity in the downstream x-direction, and a the surface mass balance term (positive for net accumulation). The velocity u is determined from stress balance (e.g., MacAyeal, 1989; Hindmarsh, 2012; Pegler, 2016),

$$2\frac{\partial}{\partial x} \left[ A^{-1/n} H \left[ \left| \frac{\partial u}{\partial x} \right|^2 + \varepsilon^2 \right]^{\frac{1-n}{2n}} \frac{\partial u}{\partial x} \right] - \tau_b$$

$$- \frac{C_w A^{-1/n}}{W^{1/n+1}} H |u|^{1/n} \frac{u}{|u|} = \rho_i g H \frac{\partial (H+b)}{\partial x},$$
(2)

with A the rate factor, n a rheological parameter,  $\varepsilon$  a parameter that regularises regions with zero strain rates (e.g., Schoof, 2006; Bueler and Brown, 2009), and  $\tau_b$  the basal shear stress. The third term parameterises the lateral shear stress with the width of the outlet glacier W;  $C_w$  is a constant (see Hindmarsh, 2012, for details).

In accordance with recent studies, we assume that the relationship between basal shear stress  $\tau_b$  and velocity u is of a regularized-Coulomb form (Zoet and Iverson, 2020; Helanow et al., 2021). For large velocities, the basal shear stress becomes independent of the velocity and only depends on the effective pressure  $N = \rho_i gH - p_w$ , the difference between ice overburden

| Name                                            | Symbol         | Value                                                          | Units                   |
|-------------------------------------------------|----------------|----------------------------------------------------------------|-------------------------|
| Rate factor                                     | A              | $1.6 \times 10^{-24}$                                          | $Pa^{-3} s^{-1}$        |
| Accumulation                                    | a              | 0.3                                                            | ${\rm m}~{\rm yr}^{-1}$ |
| Sliding parameter                               | C              | $15.8 \times 10^6$                                             | $Pa (m s^{-1})^{-1/3}$  |
| Coefficient of compressibility                  | $C_e$          | 0.0345                                                         |                         |
| Lateral drag parameter                          | $C_w$          | $2(n+1)^{1/n}$                                                 |                         |
| Void ratio at freezing                          | $\mathbf{e}_c$ | 0.5178                                                         |                         |
| Strain rate regularisation                      | $\varepsilon$  | $10^{-3}$                                                      | $yr^{-1}$               |
| Void ratio corresponding to $N=0$               | $e_0$          | 1                                                              |                         |
| Void ratio reference parameter                  | $\mathbf{e}_r$ | 0.78                                                           |                         |
| Gravitational acceleration                      | g              | 9.81                                                           | ${\rm m}~{\rm s}^{-2}$  |
| Drained hydraulically active sediment thickness | $h_0$          | 1                                                              | m                       |
| Thermal conductivity                            | k              | 2                                                              | $W\ m^{-1}\ K^{-1}$     |
| Hydraulic conductivity of the bed               | $K_d$          |                                                                | ${\rm m}~{\rm s}^{-1}$  |
| Latent heat                                     | $L_h$          | $330 \times 10^3$                                              | $\mathrm{Jkg^{-1}}$     |
| Domain length                                   | $L_x$          | $10^{6}$                                                       | m                       |
| Sliding parameter                               | m              | 1/n                                                            |                         |
| Sliding parameter                               | $\mu$          | 0.5                                                            |                         |
| Rheological parameter                           | n              | 3                                                              |                         |
| Regularization constant                         | $N_0$          | $N_r \exp\left(\frac{\mathbf{e}_r - \mathbf{e}_0}{C_e}\right)$ |                         |
| Critical effective pressure for freezing        | $N_c$          | $2 \times 10^6$                                                | Pa                      |
| Reference value of effective pressure           | $N_r$          | $10^{3}$                                                       | Pa                      |
| Geothermal heat flux                            | $q_{geo}$      | $65 \times 10^{-3}$                                            | $\rm W \ m^{-2}$        |
| Density of ice                                  | $ ho_i$        | 910                                                            | ${\rm kg}~{\rm m}^{-3}$ |
| Density of water                                | $ ho_w$        | $10^{3}$                                                       | ${\rm kg}~{\rm m}^{-3}$ |
| Melting point temperature                       | $T_m$          | 273                                                            | K                       |
| Surface temperature                             | $T_s$          | $-20 + T_m$                                                    | K                       |
| Ice stream width                                | W              | 50                                                             | km                      |
| Bed offset                                      | $z_0$          | 100                                                            | m                       |
| Bed slope                                       | $z_1$          | $10^{-3}$                                                      |                         |

**Table 1.** Model parameters and their values. Geometric  $(W, z_0, z_1)$ , environmental  $(a, T_s, A, q_{geo})$ , and basal parameters  $(h_0, C)$  were chosen to be representative of a Siple Coast ice streams and for comparability with previous studies (e.g., Robel et al., 2013, 2014; Tsai et al., 2015). Subglacial drainage parameters  $(e_r, e_c, N_r)$  are based on Tulaczyk et al. (2000a).

and water pressure. We use a regularized version of the slip law used in Tsai et al. (2015),

$$\tau_b = \frac{C|u|^m \mu N}{C|u|^m + \mu N} \frac{u}{|u|},\tag{3}$$

with m=1/n (e.g., Weertman, 1974; Pattyn et al., 2012; Brondex et al., 2017). For  $\mu N \gg C|u|^m$  this reproduces a Weertman-style sliding law  $\tau_b \sim C|u|^m$  (Weertman, 1957), while for  $\mu N \ll C|u|^m$  this mimics the Coulomb-plastic rheology  $\tau_b \sim \mu N$  expected to be applicable to West Antarctic subglacial tills (Zoet and Iverson, 2020).

We are interested in interactions of the subglacial drainage system with ice-stream flow, which are coupled through the effective pressure N. Modelling the evolution of N requires a theory for the evolution of the subglacial drainage system. We turn to this next.

# 2.2 Subglacial drainage

105

110

Many models describing subglacial drainage exist (Flowers, 2015). All models share the assumption that water is conserved at the bed and flows in the direction of hydraulic gradients (Shreve, 1972; Flowers, 2015). However, only a few models are applicable to softbedded glaciers (Walder and Fowler, 1994; Ng, 2000a, b; van der Wel et al., 2013; Bougamont et al., 2014; Bueler and van Pelt, 2015). We write conservation of subglacial water mass as (Clarke, 2005; Flowers, 2015)

$$\frac{\partial h_w}{\partial t} + \frac{\partial q_w}{\partial x} = m_b,\tag{4}$$

where  $h_w$  is the effective water depth at the bed,  $q_w$  is the water flux, and  $m_b$  is the basal melt rate.

We assume that the hydraulic conductivity only depends on the water content of the bed, as is common for distributed systems (Hewitt, 2011). This can also be modified to capture the formation of subglacial conduits (Sommers et al., 2018). However, aiming to keep our model as simple as possible, we write for the water flux

$$q_w = -\frac{K_{d,\text{eff}} h_s}{\rho_w g} \frac{\partial \Phi}{\partial x} \qquad \text{with} \quad K_{d,\text{eff}} = K_d \frac{N_c}{N}. \tag{5}$$

The effective hydraulic conductivity  $K_{d,\text{eff}}$  (in m s<sup>-1</sup>) depends on the effective pressure (e.g., Haseloff et al., 2019);  $\rho_w$  is the density of water and  $N_c$  is a constant (for convenience we use the critical effective pressure for freezing, introduced below). The hydraulic potential  $\Phi$  is given by

$$\Phi = \rho_w g z_b - N + \rho_i g H. \tag{6}$$

Its gradient drives the water flux. Equation (5) is similar to a model of distributed water flow in a system of subglacial conduits ("canals") eroded into soft beds (Walder and Fowler, 1994; Ng, 2000b).

The basal melt rate  $m_b$  satisfies an energy conservation equation,

$$m_b = \frac{1}{\rho_w L_h} \left( q_{\text{geo}} + \tau_b u + k \left. \frac{\partial T}{\partial z} \right|^+ \right), \tag{7}$$

with  $L_h$  the latent heat of fusion,  $q_{\rm geo}$  the geothermal heat flux, k the thermal conductivity of ice, and  $\partial T/\partial z|^+ \approx (T_s - T_m)/H$  the temperature gradient into the ice, which we approximate as linear between surface and melting point temperatures. This assumes that the bed is temperate; this condition is satisfied for all results shown here.

Motivated by observations of thick sediments at the base of the West Antarctic ice sheet (Alley et al., 1986; Anandakrishnan et al., 1998), we assume that water is stored locally in the till, and  $h_w$  is proportional to the void ratio e of the till and the thickness of the drained hydraulically active sediment column  $h_s$ 

$$h_w = eh_s. (8)$$

The void ratio increases with decreasing effective pressure (Tulaczyk et al., 2000a)

$$\mathbf{e} = \mathbf{e}_r - C_e \log \left( \frac{N + N_0}{N_r} \right) \tag{9}$$

where  $e_r$ ,  $C_e$ ,  $N_r$ , and  $N_0$  are constants (see table 1). This relationship is based on the empirical relationship  $e = e_r - C_e \log(N/N_r)$  between the void ratio of subglacial sediment and effective pressure (Tulaczyk et al., 2000a). The constant  $N_0 \ll N_r$  has been introduced to satisfy the boundary condition N = 0 at the grounding line (13), which now corresponds to a void ratio of  $e_0 = 1$ . Void ratios in excess of  $e_0$  would correspond to an overpressured system (N < 0), but with our choice of parameters this does not happen. We expect that additional model physics would be necessary to capture such a case.

Note that  $h_s$  is the thickness of the water-saturated sediment column if all pore space is removed from it (e = 0). As water-filled pores expand the hydraulically active sediment column, its total thickness is  $(1 + e)h_s = h_s + h_w$  (see figure 1).

# 2.3 Freezing and melting in the till

135

Both e and  $h_s$  can evolve dynamically and we require a condition in addition to (8) to determine them. This condition comes from the dynamics of freezing and melting in the sediment, which determines the thickness of the hydraulically active sediment layer  $h_s$ . We adopt the basal model of Robel et al. (2013, 2014) and Haseloff (2015),

$$e(N)\frac{\partial h_s}{\partial t} = \begin{cases} & \text{if } \left(m_b < 0 \text{ and } h_s > 0 \\ & \text{and } (N \ge N_c \text{ or } h_s \le h_0)\right) \\ \frac{e(N)}{e(N_c)}m_b & \text{if } (m_b > 0 \text{ and } 0 < h_s < h_0) \\ 0 & \text{otherwise.} \end{cases}$$

$$(10)$$

- This condition states that to overcome surface tension and extend the ice surface into the pore space between the grains (corresponding to  $h_s < h_0$ ), the effective pressure must reach a critical value  $N_c$  (see e.g., Rempel, 2008, 2009). Therefore, as long as  $N < N_c$  and  $h_s = h_0$ , the pore space expands or consolidates in response to water addition or removal. If freezing occurs while  $N < N_c$ , then water is removed from the sediment and frozen onto the base of the ice stream (figure 1a). If  $N \ge N_c$ , freezing results in a frozen fringe, i.e., ice intrudes into the void spaces of the till (figure 1b).
- In adopting (10), we make two assumptions that simplify the dynamics of basal freezing. First, once a frozen fringe has formed, any further freezing occurs inside the sediment, even if the effective pressure drops below  $N_c$ , as described by equation (10)<sub>1</sub> This assumption excludes the formation of distinct ice lenses. Second, the void ratio of the frozen sediment is always  $e_c = e(N_c)$ . It is conceivable that freezing could occur at smaller void ratios if subglacial drainage has removed water before

the onset of freezing so that  $N > N_c$ . However, melting can readily occur at effective pressures smaller than  $N_c$ , accounting for condition  $(10)_2$ .

Note that the thickness of the unexpanded, hydraulically active sediment layer  $h_s$  is bounded between 0 (completely frozen) and  $h_0$  (completely unfrozen). Our parameter choices prevent the basal sediment from freezing entirely ( $h_w = h_s = 0$ ). In this case the basal temperature could be below the melting point. This scenario might become important in some areas of Antarctica (Pattyn, 2010; Mantelli et al., 2019). We simultaneously solve for  $h_w$  and  $h_s$ , with the void ratio e determined from (8) and the effective pressure determined from (9).

## 2.4 Boundary and initial conditions

We assume an ice divide at x = 0 with zero surface slope, velocity and water flux

$$u = \frac{\partial(H+b)}{\partial x} = \frac{\partial\Phi}{\partial x} = 0 \qquad \text{at } x = 0. \tag{11}$$

The position of the grounding line is given through the floatation condition and stress balance condition

$$H = -\frac{\rho_w}{\rho_i}b$$

$$2A^{-1/n}H \left| \frac{\partial u}{\partial x} \right|^{1/n-1} \frac{\partial u}{\partial x} = \frac{1}{2}\rho_i g \left( 1 - \frac{\rho_i}{\rho_w} \right) H^2$$
at  $x = x_q$ . (12)

In the numerical model, we assume that an unbuttressed ice shelf fills the domain from the grounding line to the domain boundary, where we apply (12)<sub>2</sub>. This is mathematically equivalent to (12), which assumes that the ice sheet terminates at the grounding line (e.g., Schoof, 2007a).

Assuming ice overburden pressure equal to water pressure at the grounding line, we set the effective pressure to zero at the grounding line and assume that the till layer is completely unfrozen where it is in contact with the ocean,

$$N = 0 \quad \text{and} \quad h_s = h_0 \qquad \text{at } x = x_g. \tag{13}$$

For most cases, the model is initialised with a linearly decreasing ice thickness of  $H_{\text{initial}} = \max(2000 \text{ m} - 6 \times 10^{-3} x, 100 \text{ m})$ , a linearly decreasing effective pressure  $N_{\text{initial}} = (20.85 - 10^{-6} \frac{x}{1\text{m}})$  MPa, and  $h_s = h_0$ . To promote convergence, some runs for  $K_d \ge 3 \times 10^{-2}$  m s<sup>-1</sup> are initialised from solutions previously obtained for other values of  $K_d$ .

## 3 Results

We are primarily interested in the interaction of subglacial drainage and ice sheet flow, and therefore consider a suite of models over which hydraulic conductivity decreases from infinity to zero. We expect steady state behaviour at quasi-infinite hydraulic conductivity (e.g., Tsai et al., 2015) and oscillations between slow and fast flow at vanishing hydraulic conductivity (e.g., Robel

Figure 2. Parameter space of modelled behaviour and sample profiles. Ice is coloured according to speed, and the underlying slab of hydraulically active sediment is coloured according to effective pressure. Note that the thickness of the hydraulically active sediment is vertically exaggerated by 750% and does not account for expansion due to void-ratio changes. The red line indicates the extent of the frozen fringe into the sediment (if located at the ice-bed interface, then there is no frozen fringe). Note the upstream-travelling activation wave of fast flow in column a, the downstream-travelling activation wave in column b, and the quasi-simultaneous speed-up of the entire ice stream in column c and d. Panel e shows the steady-state solution. The displayed profiles are selected to illustrate the qualitative differences in ice-stream behaviour and match the information in figure 4. Animations of these solutions are provided as supplementary material (Haseloff et al., 2025).

et al., 2014). Our interest lies in the transition between these limiting cases. In appendix B, we non-dimensionalise the model to show that the non-dimensional group

$$\kappa = \frac{[q_w]/h_0}{[u]} = K_d \times \frac{\mu N_c}{\rho_w gaL_x} \tag{14a}$$

175 
$$\approx K_d \times 10^4 \text{ s m}^{-1}$$
 (14b)

determines the ice stream behaviour (see figure 2). It describes the ratio between the scale of water velocity  $[q_w]/h_0$  and ice velocity [u]. For  $\kappa \gg 1$ , water moves much faster through the subglacial system than the ice sheet evolves, and vice versa. Note that  $N_c$  enters (14a) through its appearance in (5), not because of freezing processes.

Ice stream behaviour spans steady streaming ( $\kappa \to \infty$ ), hydraulically controlled oscillations between fast and slow flow ( $\kappa \gg$  1), and thermally controlled oscillations ( $\kappa \ll 1$ ). These regimes are distinguished by ice stream morphology, ice discharge and, where applicable, oscillation frequencies. Notably, hydraulically controlled oscillations are characterised by a quasi-simultaneous speed-up of the entire ice stream in less than 2 yrs (figure 2d), while thermally controlled oscillations exhibit an

Figure 3. Steady-state profiles of hydraulically controlled marine ice sheets with varying values of the hydraulic conductivity of the bed  $K_d$  in (5). Top row: ice sheet profile and hydraulic potential  $\Phi$ . As in figure 2, a vertically exaggerated sediment layer is coloured according to effective pressure. Second row: basal shear stress  $\tau_b$  and effective pressure contribution  $\mu N$  to it (note that vertical axis is linear below  $\mu N_c = 1$  MPa and logarithmic above). Third row: speed. Bottom row: basal melt rate. Note that the hydraulic potential is constant for  $\kappa = 10^5$  (red line in  $a_1$ ) and gradually increases for decreasing  $\kappa$  (panels  $a_2$  and  $a_3$ ).  $\kappa = 10^2$  is the smallest value of  $\kappa$  for which steady states exist.

activation wave, which travels upstream at about 4 km/yr during the speed-up phase (figure 2a). In the regime between these two limiting cases ( $\kappa \sim 1$ ), ice stream activation can occur by downstream propagation of an activation wave at about 2 km/yr (figure 2b). We analyse the processes leading to these different regimes next, starting with the fully drained limit ( $\kappa \to \infty$ ).

### 3.1 Steady streaming

185

In the limit of quasi-infinite conductivity of the hydraulic system, water fluxes are high enough to evacuate basal melt water efficiently, so that hydraulic gradients  $(\partial \Phi/\partial x)$  cannot develop, as can be seen from rearranging (5) with (14a),

$$\frac{\partial \Phi}{\partial x} = -\frac{\mu N}{\kappa a L_x h_s} q_w \to 0 \quad \text{for} \quad \kappa \to \infty.$$
 (15)

Consequently,  $\Phi = \rho_w gb - N + \rho_i gH = \text{constant}$ . At the grounding line, boundary conditions (12) require  $\Phi(x_g) = 0$ . This requires the effective pressure to adjust such that  $N = \rho_i gH + \rho_w gb$ , that is, the subglacial water content is set by the ice

and bed geometry alone, and is independent of the basal melt rate (which is positive throughout, see figure  $3d_1$ ). This limit is also referred to as height-above-buoyancy or height-above-floatation model (Van der Veen, 1987; Asay-Davis et al., 2016; Kazmierczak et al., 2022) and it leads to high effective pressure and low water content throughout most of the domain (figure  $3b_1$ ), apart from a small region near the grounding line. Stable steady-state solutions exist, characterised by high ice thickness and surface slopes (figure  $3a_1$ ). In steady state, the water flux is given by (4), i.e.,  $q_w = \int_0^x m_b \, dx$ . Note that effective pressures in excess of the critical pressure for freezing  $N_c$  are possible, as water is removed from the sediment by subglacial drainage, rather than freezing. Here,  $\kappa \to \infty$  corresponds to  $K_d \approx 10$  m s<sup>-1</sup>.

As conductivities decrease from infinity, water is less efficiently evacuated, leading to overall higher water content and lower effective pressures, as shown in figure  $3b_{1-3}$ . However, basal shear remains high and the average water content increase from  $\bar{h}_w \approx 0.43$  m to  $\bar{h}_w \approx 0.53$  m leads to a negligible increase in ice discharge at the grounding line (figure 5c). At  $\kappa = 10^2$ , a small region of larger effective stress forms just upstream of the grounding line (figure 3b<sub>3</sub>), corresponding to a lower water content and marginally higher basal resistance. While this region of elevated effective pressure and basal shear is a subtle feature for the displayed steady state, it becomes critically important for the behaviour at smaller hydraulic conductivities, as steady states cease to exist below a critical value  $\kappa \lesssim 100$ . In this range of  $\kappa$ , basal melt evacuation cannot keep up with melt rates. This leads to a local build up in the basal water content and periodic disappearance of the region of elevated basal resistance, and hence to oscillations between fast and slow flow. We analyse the processes leading to these different dynamics next.

## 3.2 Hydraulically controlled oscillations

At moderately high hydraulic conductivity  $1 \ll \kappa \lesssim 100$ , basal melt rates are positive in all of the domain, and most of the water is continuously evacuated (figure  $4a_{3,5}$ ). However, the hydraulic conductivity is not high enough to evacuate all melt water that is produced at the ice-bed interface, and the water content of the bed slowly increases. This shrinks the region of higher basal resistance close to the grounding line (figure  $4a_4$ ). Eventually the bed is sufficiently weakened for well-lubricated conditions to exist in all of the ice stream, leading to a brief episode of fast flow, enhanced heat dissipation, and sudden grounding line advance (figure 4a at  $\approx 6600$  yrs and figure  $2d_3$ ). However, a lower effective pressure increases the effective hydraulic conductivity of the bed as  $K_{d,\text{eff}} \propto 1/N$  and during this phase more water is discharged than produced. This leads to a strengthening of the bed, eventually restoring the conditions at the beginning of the oscillation cycle. Overall, these oscillations are characterised by short peaks in ice discharge and average water content of the bed (figure  $5h_3$ -i<sub>3</sub>).

Reduction of the hydraulic conductivity decreases the period and amplitude of these oscillations (figure 5). This is due to the region of elevated effective pressure and basal shear upstream of the grounding line becoming less pronounced (compare the regions of elevated effective pressure in figure  $4a_4$  and  $b_4$ ). At smaller hydraulic conductivity, the subglacial drainage configuration becomes increasingly uncoupled from the hydraulic gradient while the importance of local melting and freezing processes increases (in fact, as the conductivity approaches zero, the subglacial water content is determined by local processes only, as discussed in section 3.4 below). The period of hydraulically controlled oscillations decreases from  $\approx 2500$  yrs at

Figure 4. Hovmöller diagrams for four different hydraulic conductivities, decreasing from top to bottom with dashed magenta line marking the grounding-line position. Column 1 shows ice thickness H, column 2 ice velocity u, column 3 basal melt rate  $m_b$ , column 4 effective pressure N, and column 5 the scaled water flux  $q_w/K_d$ . Note that the colorbar of the water flux is the inverted colorbar of the effective pressure to facilitate comparison between the two. Arrows indicate times with profiles in figure 2 and animated videos of these solutions are provided as supplementary material (Haseloff et al., 2025)

 $\kappa=50$  to a minimum of  $\approx 300$  yrs at  $\kappa=1$ ; with further reduction of  $\kappa$  the oscillation period increases again up to a period of  $\approx 800$  yrs at  $\kappa=10^{-2}$  (figure 5e).

## 3.3 Intermediate oscillations

For  $\kappa \sim 1$ , freezing becomes possible, albeit at first only in a small area close to the grounding line (figure 4b<sub>3</sub>). However, this has no noticeable effect on the effective pressure and hence basal shear stress as the water flux remains high throughout the domain (figure 4b<sub>5</sub>). Therefore, for  $\kappa = 1$ , the ice flow characteristics are similar to those of the hydraulically controlled oscillations, with the entire ice stream speeding up almost simultaneously.

For smaller  $\kappa=10^{-1}$ , freezing extends up to the ice divide and leads to strengthening of the bed, which alters the oscillation characteristics. In this case, the onset of fast flow is triggered by the thickening of ice upstream and the ensuing reduction in conductive cooling. This behaviour is shown in panels  $c_1$ – $c_5$  of figure 4 and figure 2b. At the end of the fast-flowing phase, the ice is at its thinnest, basal freezing is prevalent, and a frozen fringe forms in the downstream half of the ice stream (figure  $2b_4$  to  $b_1$ ). Gradual thickening of ice upstream leads to a transition from basal freezing to melting, decreasing the effective pressure. This leads to faster ice flow in these upstream areas, and thus the formation of a steep surface gradient at the transition between low and high effective stress. The transition between fast and slow flow travels downstream (figure  $4c_2$  and figure  $2b_1$ - $b_3$ ), similar to surge fronts in mountain glaciers (Kamb et al., 1985). While subglacial water fluxes generally mirror the inverse of the effective pressure for larger values of  $\kappa$ , here, they are elevated ahead of the downstream travelling surge front (figure  $4c_5$ ). This indicates that meltwater is transported into the region just downstream of the barrier between low and high effective stress, lowering the effective pressure ahead of the surge front, and thereby allowing its propagation.

As in the hydraulically controlled oscillations, the region of elevated effective pressure and basal shear stress downstream of the surge front acts as a buttress to the overall flow. Once this buttress vanishes, the entire ice stream speeds up, leading to grounding line advance and thinning of the ice stream. The thinner ice stream experiences larger conductive cooling rates, leading to freezing, bed strengthening, and slower flow, closing the oscillation cycle.

## 3.4 Thermally controlled oscillations

At vanishing hydraulic conductivity ( $\kappa \lesssim 10^{-2}$ ), the conditions at the bed are fully determined by the local energy balance (7),  $\partial h_w/\partial t = m_b$ . Steady states would require a locally zero basal melt rate ( $m_b = 0$ ), but this can only occur if conductive cooling exactly balances heat dissipation and geothermal heat flux everywhere. This condition is generally not satisfied. Therefore, the local effective pressure and hence basal resistance to flow change on timescales dependent on the factors controlling the basal melt rate – the heat dissipation ( $\tau_b u$  term in (7)) and the conductive cooling ( $k\partial T/\partial z|^+$  term in (7)), which in turn is controlled by the ice thickness. This leads to thermally controlled oscillations with a period of  $\approx 800$  yrs previously reported in Robel et al. (2014), see figure  $4d_1-d_5$ .

During quasi-stagnant flow phases, the ice is sufficiently thin and slow so that melt rates at the bed are negative and part of the sediment freezes. The effective pressure is at the critical effective pressure for freezing  $N_c$  in all of the domain (apart from a small boundary layer close to the grounding line which vanishes for  $\kappa \to 0$ ). Slow flow also means that less mass is

Figure 5. Bifurcation diagrams for finite  $\kappa$  (panels a-e) and selected time series after an initial spin up of 6000 years (panels f-i). Shaded area in plots a-d corresponds to the range covered during an oscillation period; points are minimum and maximum values of the oscillation. Note that hydraulically controlled oscillations are characterised by a sharp peak in ice discharge and subglacial water content (e.g., red curves in plot h and i), while thermally controlled oscillations are changing more gradually (e.g., green curve in plot h and i).

evacuated than accumulated and hence the ice sheet thickens over time. This leads to a gradual reduction of the conductive cooling; the energy balance at the bed becomes more positive and the sediment starts to thaw. Once the sediment is fully

thawed, further melt produced at the bed is stored locally, leading to a reduction of the effective pressure. As the ice is thicker towards the ice divide, this processes happens first there. However, the flow remains stabilised by the partially frozen sediment in the downstream half of the ice stream.

Eventually, the transition to the fast streaming phase is triggered at the grounding line, where an increase in ice thickness and driving stress leads to speed-up (figure 4d at 6250 years and figure  $2a_1$ - $a_2$ ). This increases the heat dissipation, lowers the basal resistance and thus triggers a positive feedback between fast flow, heat dissipation, and basal lubrication. Force balance dictates that loss of basal resistance to flow affects the flow in a small boundary layer immediately upstream through longitudinal stresses ( $\eta \partial u/\partial x$ , see e.g., Fowler and Schiavi, 1998; Robel et al., 2014). At high effective stress basal melt rates in this boundary layer are high ( $\approx 10 \text{ mm yr}^{-1}$ ), quickly lowering the effective pressure, which then speeds up the flow immediately upstream. This creates an activation wave that travels upstream, which is clearly visible in the basal melt rate (figure 4d<sub>3</sub>); its properties are discussed in detail in Robel et al. (2014). In its wake the ice stream bed is well-lubricated, leading to discharge of ice in excess of accumulation, thinning of the ice, and consequently increasing conductive cooling at the bed. This eventually leads to strengthening of the bed, slow down of flow, and transition to the stagnant phase. The frequency of these oscillations is primarily determined by how fast the ice stream can replenish its mass (i.e., the accumulation rate) and the terms in the basal melt rate (7) (i.e., k,  $T_s$  and  $q_{geo}$ , MacAyeal, 1993; Robel et al., 2013).

#### 4 Discussion

In this study, we investigate the role of an evolving subglacial drainage system on marine ice stream dynamics using the simplest model of the relevant physics. A positive feedback between fast flow and basal lubrication is incorporated through both increasing local storage of water with decreasing effective pressure and increasing drainage efficiency with decreasing effective pressure. Subglacial conditions affect marine ice sheet dynamics through the basal shear stress, which depends on the effective pressure.

We find that the efficiency of a distributed subglacial drainage system alters the mode of grounding line dynamics. Lower hydraulic conductivities result in more water storage, lower effective pressure, and consequently lower basal resistance and faster flow, i.e., generally higher slip rates, thinner ice, and less grounded ice volume (figure 5b). With lower hydraulic conductivities, the melt rate at the bed gradually drops, from positive at all points in space and time (due to thicker ice mitigating conductive cooling) to zero on average at the impermeable limit, see figure  $4a_4$ - $d_4$ . This behaviour is accompanied by a transition from a quasi-simultaneous speed-up of the entire ice stream to a gradual speed-up and slowdown (figure 5h). This speed-up can be due to downstream and upstream-travelling activation waves.

## 4.1 Ice stream oscillations and glacier surges

Observations suggest that the Siple Coast ice streams show temporal variability on timescales of 300 to 500 years (e.g., Retzlaff and Bentley, 1993; Hulbe and Fahnestock, 2007; Catania et al., 2012). While previous models have been able to reproduce stagnation and activation cycles, the period of oscillations in these models is generally on the order of 1000 yrs or longer (e.g.,

MacAyeal, 1989; Robel et al., 2013, 2014). Here, we illustrate that more frequent oscillations might be modulated by subglacial hydrology.

The similarities of the marine ice streams modelled here and the observed behaviour of surging mountain glaciers corroborate the hypothesis that the same dynamics govern both of these systems, albeit at different spatial and temporal scales (e.g., Clarke et al., 1984). In the regime intermediate between hydraulical and thermal control, the oscillations share similarities with the observed classical surges of Varigated Glacier, Trapridge Glacier and others: a downstream-travelling surge front, acceleration by up to three orders of magnitude, and a basal transition between upstream temperate conditions and downstream near-frozen bed conditions (Clarke et al., 1984; Kamb et al., 1985). Recent observations have also shown that some surges are instead characterised by activation waves travelling upglacier (Murray et al., 2003; Sevestre et al., 2018), leading to the hypothesis that there are at least two distinct surge mechanisms (Murray et al., 2003). Our results suggest that these different surge dynamics could be due to small variations in effective basal conductivity of the bed.

Previous conceptual, non-spatial ("lumped") models for surging glaciers (Fowler et al., 2001; Benn et al., 2019, 2022; Terleth et al., 2024) include the same ingredients as our model. These models show that the existence of oscillations also depends on climatic (accumulation, surface temperature) and geometric factors (bed slope). While our focus is on the dynamic behaviour resulting from variable hydraulic conductivity, it is reasonable to assume that variation of climatic and geometric factors will have an equally important effect on marine ice-stream dynamics in our model. This is supported by the appearance of the accumulation rate a and the characteristic ice sheet extent  $L_x$  in the ratio of water velocity to ice velocity  $\kappa$  (14a). Conversely, we supplement existing, zero-dimensional models for glacier surges with a spatially extended model, enabling identification and analysis of previously inaccessible complexities.

# 310 4.2 Marine ice sheet dynamics

Marine ice sheet dynamics are typically framed to be either stable or unstable, with instability equated to marine ice sheet collapse. Our results indicate that such classifications do not always apply when basal boundary conditions dynamically evolve. Instead, marine ice streams might undergo periodic cycles of retreat and re-advance. Previous studies investigating grounding line dynamics with evolving boundary conditions have also shown that thermal oscillations can temporarily stabilise grounding lines on retrograde slopes (Robel et al., 2016).

Most models simulating future ice loss under warming scenarios do not account for evolving basal boundary conditions. Our results show that basal conditions can change on decadal to centennial time scales. Faster oscillations are likely possible for different parameter choices. However, incorporation of subglacial drainage into predictive ice-sheet models is still hampered by uncertainty in how subglacial drainage should be upscaled to large spatial and temporal scales, as well as the computational cost of incorporating these dynamics.

## 4.3 Subglacial hydrology of ice streams

Given the control that effective conductivity of the bed exerts on ice stream dynamics, determining appropriate parameterizations of the effective hydraulic conductivity is crucial. The existence of subglacial till beneath parts of West Antarctica is

well established (Blankenship et al., 1986, 1987; Alley et al., 1986; Alley et al., 1987; Peters et al., 2006; Schroeder et al., 2014). Based on these observations, Walder and Fowler (1994) and Ng (1998, 2000b) propose the existence of subglacial canals partially eroded into the bed and ice. In settings typical for ice streams, the dynamics of these conduits mimic those typical for distributed subglacial systems where the effective pressure decreases with increasing water discharge. Assuming the same qualitative behaviour as Walder and Fowler (1994) and Ng (2000b), we model subglacial water flow following a Darcian-style description (e.g., Hewitt, 2011; Flowers, 2015) with  $K_{d,\text{eff}}$  the effective hydraulic conductivity. In our model,  $\kappa = 1$  corresponds to  $K_d = 3 \times 10^{-4}$  m s<sup>-1</sup> and at N = 100 kPa we get  $K_{d,\text{eff}} = K_d N_c/N = 6 \times 10^{-3}$  m s<sup>-1</sup>. This is consistent with in-situ measurements of the hydraulic conductivity at the base of Whillans Ice stream, where a minimum of  $K = (0.5 \text{ to } 1.4) \times 10^{-3}$  m s<sup>-1</sup> has been estimated (Engelhardt and Kamb, 1997).

However, such high hydraulic conductivities are inconsistent with subglacial water flow within the void space of the till alone. For example, for subglacial tills derived from Whillans ice stream, Leeman et al. (2016) find conductivities around  $10^{-12}$  m s<sup>-1</sup>. Measurements in the field report values around  $2 \times 10^{-9}$  m s<sup>-1</sup> (Engelhardt et al., 1990). Moreover, the Kozeny–Carman relationship  $K \propto e^3/(1+e)$  is typically used for flow of water in the void space e of sediment (e.g., Lambe and Whitman, 1991). When combined with the measured dependence of the void ratio on effective pressure (9), a different dependence of K on K arises than what we assumed here. This might also alter the ice stream dynamics. In our hydraulically-controlled oscillations, the increase of drainage efficiency with lower K causes the termination of the fast flow phase, as more water is evacuated as the bed becomes more lubricated. In the Kozeny–Carman relationship, the dependence on K is weaker than in our model, which might not permit the dynamics seen here.

Assuming that intermediate  $\kappa$  are representative for West Antarctic ice stream dynamics, alternative pathways must facilitate a more efficient water transport. Engelhardt and Kamb (1997) suggest that a distributed system of sediment-incised canals, as proposed by Walder and Fowler (1994) is most compatible with their observations. Between these canals, water might flow in a thin film at the ice bed-interface, rather than in the subglacial sediment, further enhancing the hydraulic conductivity (Creyts and Schoof, 2009).

For high water discharge conditions, it is plausible that a transition from distributed to channelised drainage might occur (Walder and Fowler, 1994; Röthlisberger, 1972; Hager et al., 2022; Dow et al., 2022). Under these conditions, we expect the net discharge to increase with effective pressure, rather than decrease. In Greenland, where abundant surface meltwater can drain to the bed, increasing channelisation leads to a strengthening of the bed at the end of the melt season (Schoof, 2010; Sundal et al., 2011; Bartholomew et al., 2012). We also expect the conductivity to vary locally, so that we interpret  $K_{d,eff}$  as an effective hydraulic conductivity, averaged over spatial scales resolvable in ice sheet models.

We have assumed that the basal shear stress adjusts instantaneously to changes in subglacial water content, as we are interested in the annual to centennial evolution of marine ice sheets. However, at the grounding line, marine ice sheets are subject to tidal forcing on much shorter timescales. At such short timescales, transient dilatant strengthening of subglacial till can lead to time-dependent sliding laws and affect grounding line motion (Warburton et al., 2020, 2023). It has been hypothesised that time-dependent sliding laws might also modulate surging of glaciers (Minchew and Meyer, 2020); this behaviour might be

important for the quasi-simultaneous speed-up we observed in the hydraulically controlled limit at moderately high hydraulic conductivities.

## 360 4.4 Model limitations

In the formulation of our model, we have made simplifying assumptions that limit its applicability. Most notably, we use a width-integrated model with a parameterized lateral drag term. Assuming that the width of the ice stream is constant allows us to ignore the complex dynamics required to capture its evolution. However, observations indicate that ice-stream widths can change over time (e.g., Harrison et al., 1998; Echelmeyer and Harrison, 1999). Modelling also shows that substantial englacial melt is produced in ice-stream margins, which could affect ice stream hydrology and dynamics (Raymond, 2000; Suckale et al., 2014; Haseloff et al., 2019). Future modelling efforts should build on our work to account for the hydraulic and thermal processes affecting ice stream width.

Another limitation of the width-averaged approach is that it precludes the formation of ice ridges, whose surface slopes affect the ice stream stability by altering the hydraulic gradient (Kyrke-Smith et al., 2014) and the interaction of neighbouring ice streams. For example, it has been hypothesised that changes in subglacial water pathways might have contributed to the shutdown of Kamb ice stream approximately 140 years ago (e.g., Anandakrishnan and Alley, 1997). The advantage is that we are left with a minimal model of a marine ice stream in which dissipation, drainage, and freezing processes compete to shape the ice stream thickness and velocity field, and we can study the effect of different physical processes in the model.

We also ignore the role of ice shelves here. Buttressing ice shelves can alter marine ice stream dynamics (Gudmundsson et al., 2012; Gudmundsson, 2013; Haseloff and Sergienko, 2018, 2022). In addition, at the grounding line, subglacial hydrology can interact with the ocean in multiple ways not taken into account here, for example through seawater infiltration and the initialisation of subshelf meltwater plumes which might alter ice shelf buttressing (e.g., Jenkins, 2011; Robel et al., 2022; Ehrenfeucht et al., 2024). More generally, where marine ice streams are in contact with stochastically-varying ocean and atmospheric conditions, their dynamics can significantly diverge from purely deterministic results (e.g., Mantelli et al., 2016; Robel et al., 2018; Christian et al., 2022; Sergienko and Haseloff, 2023). Such effects should be addressed in future studies.

Another model limitation is that we do not allow for full freezing of the bed; in all simulations reported here the bed remains temperate. Observations suggest temperate ice stream beds (e.g., Engelhardt and Kamb, 1997) but the bed might be frozen under the adjacent ice ridges (e.g., Bentley et al., 1998). Frozen–temperate transitions might also play a role in ice stream onset (Mantelli et al., 2019), although in our model upstream regions usually have higher water content, as the thicker ice there reduces conductive cooling of the bed. The formation of temperate to frozen boundaries is not possible in our model, and additional model physics would be required to capture these (Haseloff et al., 2015, 2018; Mantelli and Schoof, 2019; Schoof and Mantelli, 2021).

## 5 Conclusions

In this paper we showed that subglacial hydrology affects the stability, extent, and grounded ice volume of marine ice streams.

Different dynamical regimes, including steady streaming, hydraulically controlled, quasi-simultaneous advances, and periodic oscillations with advancing or retreating activation fronts are possible for different hydraulic conductivities of the bed. Our results illustrate that marine ice streams can undergo internal variability at periods from a few centuries to millennia. This variability may need to be taken into account for accurate prediction of future sea level contributions on these timescales.

Code availability. The code used in this study is available on Zenodo (Haseloff, 2025).

Video supplement. https://doi.org/10.5446/s\_1864 (Haseloff et al., 2025)

# Appendix A: Model implementation and verification

The model is implemented in PETSc (Balay et al., 2023). PETSc's SNES library for solution of nonlinear systems of algebraic equations is used to solve individual equations. The equation for momentum balance (2) and subglacial drainage (4)–(9) are discretised using conservative finite differences with implicit timesteps (Katz et al., 2007). The equation for mass balance (1) is discretised with a third-order upwind scheme. We also use a linear subgrid interpolation at the grounding line (Pattyn, 2003).

At each timestep, we use a segregated loop to solve our model. That is, the equations for mass balance (H), momentum balance (u), and subglacial water content  $(h_s$  and  $h_w)$  are solved in three independent steps. To ensure that the combined solution at the current timestep is converged, the steps are iterated using a Picard scheme until convergence is achieved. That is, at each timestep  $t_i$  we calculate  $u(t_i^1)$ ,  $H(t_i^1,x)$ ,  $h_s(t_i^1,x)$  and  $h_w(t_i^1,x)$  in the first iteration and use these to continue to calculate  $u(t_i^2,x)$ ,  $H(t_i^2,x)$ , etc until further iteration does not alter the solution at the current timestep. The size of the timestep is adjusted to achieve convergence of the individual SNES solvers as well as convergence of the segegrated solution in no more than 10 Picard iterations.

The implemented scheme is of  $O(\Delta x)$  accuracy, and comparison of steady-state results and timeseries under grid refinement are shown in figure A1. While the qualitative behaviour is consistent for values of  $\Delta x \lesssim 1000$  m, convergence requires resolution at 10s of meters. Solutions shown here were calculated at resolutions of  $\Delta x = 100$  m or finer. Simulation times depend on the hydraulic conductivity parameter and of course model resolution. For  $\kappa \lesssim 1$  and a grid spacing of  $\Delta x = 100$  m,  $10^4$  model years are computed in about 24 hours on a single core. Simulations with  $\kappa \gg 1$  require significantly more time (up to two weeks) due to limitations on the timestep.

Figure A1. Numerical convergence study. Panels a-b show grounding line position  $x_g$  and average water content  $\bar{h}_w$  under grid refinement for steady-state solutions ( $\kappa = \infty$ ), panels c-d show timeseries of the same for oscillating solutions ( $\kappa = 10$ ).

# Appendix B: Non-dimensional model

Here, we non-dimensionalize equations (1)–(4) to highlight scaling behaviour in each balance region. There are two limiting cases. For hydraulically controlled ice streams, ice is relatively thick, velocities are slow, and basal resistance is high, with both the Weertman-like terms and the Coulomb-like terms in equation (3) contributing to the basal shear stress. Subglacial water flow is efficient and the sediment does not experience freezing. We consider this limit first in section B1. For thermally controlled ice streams, ice is thin and basal resistance is primarily of the Coulomb-plastic type. The basal energy balance oscillates between melting and freezing; this limit is considered below in section B2.

## **B1** Hydraulically controlled ice streams

In this limit, we choose the following scales with characteristic values as indicated,

$$[x] = L_x = 1000 \text{ km},$$
 (B1a)

$$[u] = \left(\frac{\rho_i g a^2[x]}{C}\right)^{n/(2n+1)} \approx 60 \text{ m yr}^{-1}, \tag{B1b}$$

$$[H] = \frac{a[x]}{[u]} \approx 4700 \text{ m},$$
 (B1c)

$$[t] = \frac{[H]}{a} \approx 15 \times 10^3 \text{ yr},\tag{B1d}$$

$$[N] = \frac{1}{\mu} C[u]^{1/n} \approx 400 \text{ kPa.}$$
 (B1e)

We set  $x = [x]x^*$ ,  $H = [H]H^*$ ,  $u = [u]u^*$ ,  $t = [t]t^*$ ,  $N = [N]N^*$ ,  $h_s = h_0h_s^*$ , and  $\Phi = \rho_i g[H]\Phi^*$ . Dropping asterisks immediately, we obtain the non-dimensional equations for momentum balance (2), mass balance (1), and water balance (4),

$$2\alpha_{1} \frac{\partial}{\partial x} \left( H \left| \frac{\partial u}{\partial x} \right|^{1/n - 1} \frac{\partial u}{\partial x} \right) - \alpha_{2} H u^{1/n} - \frac{N u^{1/n}}{u^{1/n} + N} \frac{u}{|u|} = H \frac{\partial H}{\partial x}$$
(B2a)

$$\frac{\partial H}{\partial t} + \frac{\partial (uH)}{\partial x} = 1 \tag{B2b}$$

$$\frac{\partial(eh_s)}{\partial t} + \kappa \frac{\partial}{\partial x} \left( \frac{1}{N} \frac{\partial \Phi}{\partial x} \right) 
= \left( \alpha_3 + \alpha_4 \frac{Nu^{1/n+1}}{u^{1/n} + N} - \alpha_5 \frac{1}{H} \right)$$
(B2c)

with the non-dimensional groups

$$\alpha_1 = \frac{[H]}{A^{1/n}CL_x^{1/n+1}} \approx 3 \times 10^{-4},\tag{B3a}$$

$$\alpha_2 = \frac{C_w[H]}{A^{1/n}CW^{1/n+1}} \approx 4 \times 10^{-2},$$
 (B3b)

$$\kappa = K_d \frac{\mu N_c}{\rho_w g L_x a} \approx K_d \times 10^4 \text{ s m}^{-1}, \tag{B3c}$$

$$\alpha_3 = \frac{q_{\text{geo}}[H]}{\rho_w L_h h_0 a} \approx 100,\tag{B3d}$$

$$\alpha_4 = \frac{C[u]^{1/n}[x]}{\rho_w L_h h_0} \approx 600,$$
(B3e)

$$\alpha_5 = \frac{k\Delta T}{\rho_w L_h h_0 a} \approx 13. \tag{B3f}$$

With  $\alpha_1 \ll \alpha_2 \ll 1$ , the leading order momentum balance is between basal resistance and driving stress,

$$-\frac{Nu^{1/n}}{u^{1/n}+N}\frac{u}{|u|} \sim H\frac{\partial H}{\partial x}.$$
(B4)

In the basal water balance, the non-dimensional group of interest is  $\kappa$ , which controls the importance of subglacial hydrology. It can be understood as the ratio of subglacial water velocity to ice velocity, as

$$\kappa = \frac{[q_w]/h_0}{[u]} \quad \text{with} \quad [q_w] = K_d \frac{N_c h_0[\Phi]}{\rho_w g[N][x]}. \tag{B5}$$

If  $K_d \sim (\rho_w g L_x a)/(\mu N_c)$ , then  $\kappa \sim 1$ . We also note that  $\alpha_5 \ll \alpha_4$ , i.e., freezing is negligible.

For  $\kappa \to \infty$ , we expect steady-states with constant hydraulic potential as detailed in (15). Practically, this limit is attained for  $K_d \sim 10 \text{ m s}^{-1}$  (corresponding to  $\kappa \approx 10^5$ ). As long as  $\kappa \gg 1$ , we expect subglacial hydrology to be efficient; we call this limit hydraulically controlled. In this case, the leading order water balance is

$$\kappa \frac{\partial}{\partial x} \left( \frac{1}{N} \frac{\partial \Phi}{\partial x} \right) \sim \alpha_4 \frac{N u^{1/n+1}}{u^{1/n} + N}. \tag{B6}$$

For  $\kappa \ll 1$  we expect subglacial hydrology to become increasingly inefficient and the interplay between local melt production and freezing to dominate the basal energy balance. This requires thinner ice streams; we rescale below to capture this asymptotic limit.

The limit  $\kappa \sim 1$  describes the intermediate regime. In this limit, the relative magnitudes of subglacial water flux, geothermal heat flux, basal energy dissipation, and conductive cooling determine the energy balance.

# 455 B2 Thermally controlled ice streams

In this limit, we choose the following scales

$$[x] = L_x = 1000 \text{ km},$$
 (B7a)

$$[H] = \frac{k\Delta T}{q_{\text{eeo}}} \approx 600 \text{ m}, \tag{B7b}$$

$$[u] = \frac{a[x]}{[H]} \approx 500 \text{ m yr}^{-1},$$
 (B7c)

$$[t] = \frac{[H]}{a} \approx 2000 \text{ yr},$$
 (B7d)

$$[N] = \frac{1}{\mu} \rho_i g \frac{[H]^2}{[x]} \approx 7 \text{ kPa.}$$
 (B7e)

We set  $x=[x]x^{\dagger}$ ,  $H=[H]H^{\dagger}$ ,  $u=[u]u^{\dagger}$ ,  $t=[t]t^{\dagger}$ ,  $N=[N]N^{\dagger}$ ,  $h_s=h_0h_s^*$ , and  $\Phi=\rho_ig[H]\Phi^{\dagger}$ . Dropping daggers immediately, we obtain the non-dimensional equations for mass balance (1), momentum balance (2), and water balance (4),

$$2\gamma_{1} \frac{\partial}{\partial x} \left( H \left| \frac{\partial u}{\partial x} \right|^{1/n - 1} \frac{\partial u}{\partial x} \right) - \gamma_{2} H u^{1/n}$$

$$- \frac{\gamma_{3} N u^{1/n}}{\gamma_{3} u^{1/n} + N} \frac{u}{|u|} = H \frac{\partial H}{\partial x}$$
(B8a)

$$465 \quad \frac{\partial H}{\partial t} + \frac{\partial (uH)}{\partial x} = 1 \tag{B8b}$$

$$\frac{\partial (eh_s)}{\partial t} + \kappa \frac{\partial}{\partial x} \left( \frac{1}{N} \frac{\partial \Phi}{\partial x} \right) 
= \gamma_4 \left( 1 - \frac{1}{H} \right) + \gamma_5 \frac{\gamma_3 N u^{1/n+1}}{\gamma_3 u^{1/n} + N}$$
(B8c)

with non-dimensional groups

$$\gamma_1 = \frac{a^{1/n}}{\rho_i g A^{1/n} [H]^{1/n+1}} \approx 4 \times 10^{-3},\tag{B9a}$$

$$\gamma_2 = C_w \left(\frac{[x]}{W}\right)^{1/n+1} \gamma_1 \approx 0.7,\tag{B9b}$$

$$\gamma_3 = \frac{Ca^{1/n}[x]^{1/n+1}}{\mu \rho_i g[H]^{2+1/n}} \approx 120,$$
 (B9c)

$$\gamma_4 = \frac{1}{\rho_w L_h} \frac{k\Delta T}{ah_0} \approx 13,\tag{B9d}$$

$$\gamma_5 = \frac{\rho_i g}{\rho_m L_b h_0} \left(\frac{k\Delta T}{q_{\text{geo}}}\right)^2 \approx 10.$$
 (B9e)

As before, the dimensional group determining the role of subglacial hydrology is  $\kappa$ . Subglacial water flux is negligible for  $\kappa \ll 1$  which describes thermally controlled oscillations. In this limit the leading order water balance (neglecting terms of  $O(1/\gamma_3)$ ) is

$$\frac{\partial(eh_s)}{\partial t} \sim \gamma_4 \left(1 - \frac{1}{H}\right) + \gamma_5 Nu. \tag{B10}$$

This limit is attained for  $K_d \lesssim 10^{-6} \text{ m s}^{-1}$ .

Author contributions. MH designed the study, formulated the model, performed the research, and wrote the manuscript. IJH advised on the model formulation and interpretation. RFK advised on the numerical solution of the model. All authors secured funding for part of the project
 and contributed to the editing of the manuscript.

Competing interests. The authors declare no competing interests.

Acknowledgements. This study was supported by award NSF2218463 from the National Science Foundation and NERC Grant NE/R000026/1. We thank the reviewers Alexander Robel and Elise Kazmierczak and the editor Johannes Sutter for their reviews of our manuscript. Computations were performed at the Center for High Throughput Computing (2006).

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
