# Peer review of "Subglacial hydrology regulates oscillations in marine ice streams"

_EGUsphere, 2025_

## Referee Comment (RC2)

The paper is well-written and demonstrates a high level of scientific quality. The topic is of significant interest and aligns well with current research on the influence of subglacial hydrology on ice motion, particularly within the context of the Antarctic ice sheet. This area of study has gained renewed attention in recent years, addressing gaps in the existing literature.

The research demonstrates the impact of subglacial hydrology on marine ice stream dynamics, coupling various systems in a positive feedback loop involving ice flow, basal heat dissipation, and basal lubrication. By using a simple flow-line model, the authors explore different subglacial phenomena arising from the presence of subglacial water in a soft bed system. Their results emphasize the dependence of ice stream dynamics on basal conditions.

I recommend the publication of this paper following a few minor revisions.

I have structured the comments into general comments and line-by-line comments. The general comments are suggestions to improve clarity for a new reader, curiosity questions, or suggestions of improvements that are recurrent throughout the paper.

**General comments and suggestions**

- I suggest to summarize in a simple and clear way the model at the beginning of the paper. Indeed, it is a bit complicated during the first read of the paper to understand well in terms of subglacial hydrology what is considered and what is not. For example, please specify that it's a model only applied on a soft bed and which encompasses inefficient and efficient subglacial drainage systems through a hydraulic conductivity factor that increases with decrease of the effective pressure. No switch between efficient and inefficient drainages is assumed, as well as no channels at higher subglacial water flux. Furthermore, no interaction between subglacial water and ocean water and no vertical infiltration are considered. I also think you could more explicitly summarize what subglacial processes the kappa factor encompass.

- Also, I do not understand whether or not $h_{w\&s}$ are limited. That is the case, please mention it.

- Some values used in this research (e.g., constant coefficients in Table 1 or values of kappa used in experiments) are not referenced, and there is no explanation regarding the choice of their values. Please provide the references where applicable and add comments about the choices that had to be made.

- I think it's important to clearly specify that the processes you talk about are subglacial processes. Therefore, make sure to add the term "basal" with "melt" and "subglacial"

with "water" when necessary. The same goes for the term "discharge"—make sure to specify whether it refers to water or ice.

- In general, in the presentation of the results, I suggest emphasizing the quantitative values of the mentioned variations and referencing the relevant figures/videos more regularly.

- Does one of your limit cases could be interpreted as that of a hard bed system ?

- Have other geometries (slope/shape) of beds been investigated?

- In terms of results, more details could be provided regarding the timing of the experiments. For example, I don't understand why, in Figure 2, there is a 800-year difference between d1 and d2, while in the other figures, the results are shown to take place over a scale of a few years. I also think a comment on the initial and final conditions would be interesting.

- What is the computation time for these models?

- In your text, you explain well that kappa is an average used to represent in 1D hydrological systems that would be found in 2D. You also mention that multiple systems can exist for N. However, depending on the drainage system, for the same flow, a different effective pressure could be obtained (cf. Fig. 4 Walder and Fowler, 1994). Maybe you discuss how your model could include other drainage systems.

**Line-by-line comments**

L6: I don't understand why a hydraulically controlled motion is called a "**surge**", while a thermally controlled motion is referred to as an "**oscillation**". Shouldn't we use the same term, or are these two different phenomena?

L25: Maybe add something that explains why existing observational work does not address the interplay between subglacial hydrology and marine ice-sheet dynamics, such as 'because of the lack of direct observation.'

L27: Maybe add: Gregov, T., Pattyn, F., & Arnst, M. (2023). Grounding-line flux conditions for marine ice-sheet systems under effective-pressure-dependent and hybrid friction laws. *Journal of Fluid Mechanics*, *975*, A6.

L40: By reading this sentence, one might think that in an extreme case, this feedback loop never stops. Is it possible to add something to moderate it, like "up to a certain point"?

L44: add something like "Which is often the case because of the low porosity of this material"

L52: "**subglacial** water" – "and water content of the bed **composed by till**"

L63: Give a numerical value for the low hydraulic conductivity

L64: At what value of hydraulic conductivity is the limiting case obtained?

L68: finite **and intermediate** hydraulic conductivities

**2 The Model**

L76: It's not clear that the length refers to the domain and not to the flowline.

L77: **depth** or **thickness** to clarify the word "content"

L86-87: For $n$, keep "rheological" as used in the text or "viscosity" as in the table, but not both—avoid using two different terms. The value of epsilon is missing in the table. What is the value, unit, and source of epsilon? Also, please provide more details about $C_w$.

L88: Refer to some of these recent studies.

L92: Add a note that explains that since $\mu$ **is constant** at high effective pressure, you obtain a Weertman-style sliding law. Also, include a reference explaining why **m = 1/3** is used.

L99: Maybe add the reference: Shreve, R. L. (1972). Movement of water in glaciers. *Journal of Glaciology*, *11*(62), 205-214.

L100: Maybe add references like van der Wel et al., 2013, Bougamont et al., 2014 and Bueler and van Pelt, 2015.

van der Wel, N., Christoffersen, P., & Bougamont, M. (2013). The influence of subglacial hydrology on the flow of Kamb Ice Stream, West Antarctica. *Journal of Geophysical Research: Earth Surface, 118*(1), 97-110.

Bougamont, M., Christoffersen, P., Hubbard, A. L., Fitzpatrick, A. A., Doyle, S. H., & Carter, S. P. (2014). Sensitive response of the Greenland Ice Sheet to surface melt drainage over a soft bed. *Nature communications*, *5*(1), 5052.

Bueler, E., & van Pelt, W. (2015). Mass-conserving subglacial hydrology in the Parallel Ice Sheet Model version 0.6. *Geoscientific Model Development*, *8*(6), 1613-1635.

L100-L102: **subglacial** water

L103: If I understand correctly, the properties of the bed itself are not modified. I would suggest modifying the sentence to: "The conductivity only depends on the water content […] and the properties of the bed are kept constant".

L109: The difference of pressures allows canals to deform the soft bed. So I will modify the sentence by "eroded **and deformed**".

L120: For the values of the model-specific constants, I suggest adding the sources of the values used.

L140: Add a reference for this.

L14 : Add numbers/letters in Eq. 10 conditions for more clarity. I propose 10b.

L153: Maybe add that the GL and the calving front are the same (even though it is mentioned on line 336).

**3 Results**

L174: **basal** melt water

L 175: In the limiting case of quasi-infinite conductivity of the hydraulic system, I do not understand how water can be evacuated without hydraulic gradients. From my understanding, this limit case corresponds to the height above buoyancy model. Could you provide me with further explanations?

L 185: you mention fig 5 before fig 4 in the text

L 199: please refer to the corresponding figure to find directly the order of magnitude of variations and the time scale

L209: Figure 5**e**

L222: **subglacial** water fluxes

L232-238: **basal** melt rate

L238-235: Please add numerical values to your analysis.

L250-256: Also, remind us which figure we should observe.

L250: "significant" – Maybe provide a numerical value, if one is available?

**4 Discussion**

L261: I would nuance this statement by adding "**namely**"

L263: specify the kind of subglacial drainage system

L264: maybe remind the reader that low hydraulic conductivities lead to high subglacial water storage and reduce effective pressure

L288: indic**a**te

L299: Do you obtain such a result also because you considered more things in your "hydraulic conductivity parameter"?

L306: maybe add reference like Schroeder, D. M., Blankenship, D. D., Young, D. A., Witus, A. E., & Anderson, J. B. (2014). Airborne radar sounding evidence for deformable sediments and outcropping bedrock beneath Thwaites Glacier, West Antarctica. *Geophysical Research Letters*, *41*(20), 7200-7208.

L307: and deforming the bed composed of till/sediments

L313: Papers like Hager 2022 and Dow 2022 consider channels in WAIS. Maybe add a sentence assuming that their existence is plausible in WAIS too.

Hager, A. O., Hoffman, M. J., Price, S. F., & Schroeder, D. M. (2022). Persistent, extensive channelized drainage modeled beneath Thwaites Glacier, West Antarctica. *The Cryosphere*, *16*(9), 3575-3599.

Dow, C. F. (2022). Hidden rivers under Antarctica impact ice flow and stability. *nature geoscience*, *15*, 869-870.

**Appendix**

L395: Inefficient rather than ineffective (by opposition to "efficient" in line 393).

**Figures and table**

General comments for the figures : Great figures! However, I always find it clearer when the extreme values of the colorbar are also indicated. Verify that all panels are mentioned and explained in the short text linked to each figure.

Figure 1: not clear that the bed is "downwards"

Table 1: If you don't explain all the parameters in the text, refer to the table. Epsilon, $h_m$, $N_0$ are missing from it.

Figure 2: Explain column c and e.

Figure 3: Why showing specifically these values of kappa ? Maybe provide a rationale for that choice.

Figure 4: **basal** melt rate

Figure 5: **Ice discharge / subglacial** water

---

## Author Comment (AC1)

**Response to reviewer 1 (Alexander Robel)**

M. Haseloff, I. J. Hewitt, R. F. Katz

May 19, 2025

*We thank the reviewer for their thoughtful and constructive review. In this document, reviewer text is in black and our response to it is in blue. Manuscript text is in italics, with original text in black and changes in red.*

This is a review of the manuscript "Subglacial hydrology regulates oscillations in marine ice streams" by Haseloff et al. for publication in The Cryosphere, prepared by Alexander Robel. This paper describes a 1D model and corresponding results on the coupled variability of marine ice stream flow and subglacial hydrology. It extends prior work by myself and others, considering water flow within subglacial hydrological systems which leads to the emergence of more rapid surge-like ice flow variability in realistic parameter regimes that may help to address certain shortcomings of past models. I think the manuscript is overall well written and the scientific results are robust and thoughtfully laid out. I see no barrier to publication of this in The Cryosphere after some minor revisions.

**Conceptual suggestions:**

a) I think perhaps one of the more underplayed implications of the results is that the model produces variability at multi-centennial time scales at values of till hydraulic conductivity that may reasonably be expected to occur in reality. Prior models (i.e., Robel et al. 2013, 2014) which do not include subglacial water flow cannot produce ice stream flow oscillations on time scales less than 900 years or so. The best observations we have of ice stream flow variability in the present day (or at least the late holocene) is the Siple Coast, where most evidence points to periodicity in stagnation-activation cycles of 300-500 years. So, this is a very exciting result since it may resolve this issue. It would be worth spending more time on this in the discussion, and also comparing to the results of Mantelli et al. 2016 which is able to produce quasi-periodic variability at similar time-scales by forcing the system with stochastic climate noise.

*The following paragraph has been added to the discussion about ice stream variability, section 4.1:*

*Observations suggest that the Siple Coast ice streams show temporal variability on timescales of 300 to 500 years (e.g., Retzlaff and Bentley, 1993; Hulbe and Fahnestock, 2007; Catania et al., 2012). While previous models have been able to reproduce stagnation and activation cycles, the period of oscillations in these models is generally on the order of 1000 yrs or longer (e.g., MacAyeal, 1989; Robel et al., 2013, 2014). Here, we illustrate that more frequent oscillations might be modulated by subglacial hydrology.*

*The following has been added to the discussion about model limitations, section 4.4:*

*More generally, where marine ice streams are in contact with stochastically-varying ocean and atmospheric conditions, their dynamics can significantly diverge from purely deterministic results (e.g., Mantelli et al., 2016; Robel et al., 2018; Christian et al., 2022; Sergienko and Haseloff, 2023).*

b) I think you a being too non-committal on the question of what $K_d$ is in reality. On line 298, you say "these values are not straightforwardly transferable to the values of $K_d$ used here. These values do not take the formation of subglacial conduits into account." But you basically have reasonable values for the case where flow occurs entirely through microporous till ($K$ small limit)

and where channelization may enhance these values ($K$ intermediate to large). Other studies (Warburton et al., 2020) besides those that you have cited indicate that till under shear in W. Antarctica may have these higher conductivities (without specific evidence for channelization). You should translate these to Kappa in the discussion in 4.3 so readers can understand what these till values mean in the context of your prior results. Ultimately, the strength of these results will rest on whether the reader understands the conditions under which they can be applied to reality, even if uncertainty remains in what exact parameter values are.

Excellent point. We have rewritten the discussion on ice stream hydrology in the following way:

*Given the control that effective conductivity of the bed exerts on ice stream dynamics, determining appropriate parameterizations of the effective hydraulic conductivity is crucial. The existence of subglacial till beneath parts of West Antarctica is well established (Blankenship et al., 1986, 1987; Alley et al., 1986; Alley et al., 1987; Peters et al., 2006; Schroeder et al., 2014). Based on these observations, Walder and Fowler (1994) and Ng (2000) propose the existence of subglacial canals partially eroded into the bed and ice. In settings typical for ice streams, the dynamics of these conduits mimic those typical for distributed subglacial systems where the effective pressure decreases with increasing water discharge.* ~~Estimates of hydraulic conductivity of till vary between $10^{-9}$ to $5 \times 10^{-5}$ m s$^{-1}$ for different locations (Fountain and Walder, 1998). However, these values are not straightforwardly transferable to the values of $K_d$ used here. These values do not take the formation of subglacial conduits into account. Such conduits likely increase the effective hydraulic conductivity and introduce a dependence on the effective pressure. For example, slug tests have yielded enhanced hydraulic conductivities of unconsolidated sediments close to a subglacial channel (from $10^{-3}$ to $10^{-2}$ m s$^{-1}$, Kulessa et al., 2005).~~ *Assuming the same qualitative behaviour as Walder and Fowler (1994), we model subglacial water flow following a Darcian-style description (e.g., Hewitt, 2011; Flowers, 2015) with $K_{d,eff}$ the effective hydraulic conductivity. In our model, $\kappa = 1$ corresponds to $K_d = 3 \times 10^{-4}$ m s$^{-1}$ and at $N = 100$ kPa we get $K_{d,eff} = K_d N_c/N = 6 \times 10^{-3}$ m s$^{-1}$. This is consistent with in-situ measurements of the hydraulic conductivity at the base of Whillans Ice stream, where a minimum of $K = (0.5$ to $1.4) \times 10^{-3}$ m s$^{-1}$ has been estimated (Engelhardt and Kamb, 1997).*

*However, such high hydraulic conductivities are inconsistent with subglacial water flow within the void space of the till alone. For example, for subglacial tills derived from Whillans ice stream, Leeman et al. (2016) find conductivities around $10^{-12}$ m s$^{-1}$. Measurements in the field report values around $2 \times 10^{-9}$ m s$^{-1}$ (Engelhardt et al., 1990). Moreover, the Kozeny–Carman relationship $K \propto e^3/(1 + e)$ is typically used for flow of water in the void space $e$ of sediment (e.g., Lambe and Whitman, 1991). When combined with the measured dependence of the void ratio on effective pressure (9), a different dependence of $K$ on $N$ arises than what we assumed here. This might also alter the ice stream dynamics. In our hydraulically-controlled oscillations, the increase of drainage efficiency with lower $N$ causes the termination of the fast flow phase, as more water is evacuated as the bed becomes more lubricated. In the Kozeny–Carman relationship, the dependence on $N$ is weaker than in our model, which might not permit the dynamics seen here.*

*Assuming that intermediate $\kappa$ are representative for West Antarctic ice stream dynamics, alternative pathways must facilitate a more efficient water transport. Engelhardt and Kamb (1997) suggest that a distributed system of sediment-incised canals, as proposed by Walder and Fowler (1994) is most compatible with their observations. Between these canals, water might flow in a thin film at the ice bed-interface, rather than in the subglacial sediment, further enhancing the hydraulic conductivity (Creyts and Schoof, 2009).*

~~More generally, modelling West Antarctic basal environments requires models for water flow on soft (permeable and deformable) beds. Most subglacial drainage models are developed for hard (impermeable and undeformable) beds, but the existence of subglacial till beneath parts of West Antarctica is well established (Blankenship et al., 1986, 1987; Alley et al., 1986; Alley et al., 1987; Peters et al., 2006; Schroeder et al., 2014). Based on these observations, Walder and Fowler (1994) and Ng (2000) propose the existence of subglacial canals partially eroded into the~~

~~bed and ice. In settings typical for ice streams, the dynamics of these conduits mimic those typical for distributed subglacial systems where the effective pressure decreases with increasing water discharge. Assuming the same qualitative behaviour as Walder and Fowler (1994), we have used $K_{d,eff} \propto 1/N$. However, a more complicated relationship is not only possible but likely for realistic ice streams, where a range of subglacial drainage systems might exist at the same value of N (Flowers, 2015). In particular,~~

*For high water discharge conditions, it is plausible that a transition from distributed to channelised drainage might occur (Walder and Fowler, 1994; Röthlisberger, 1972; Hager et al., 2022; Dow et al., 2022). Under these conditions, we expect the net discharge to increase with effective pressure, rather than decreases.  In Greenland, where abundant surface meltwater can drain to the bed, increasing channelisation leads to a strengthening of the bed at the end of the melt season (Schoof, 2010; Bartholomew et al., 2012). We  also expect the conductivity to vary locally, so that we interpret $K_{d,eff}$ as an effective hydraulic conductivity, averaged over spatial scales resolvable in ice sheet models.*

c) I see that there is some discussion of the numerics in the appendix, but it would be useful to summarize in the model description sections how the equations are discretized and solved (seemingly some large nonlinear solve in Petsc). Particularly because activation/surge-type behavior in models is notoriously resolution dependent and you do reference the computational intensity of these simulations in your discussion.

The section describing the model has been extended in the following way:

*The model is implemented in PETSc (Balay et al., 2023) PETSc's SNES library for solution of nonlinear systems of algebraic equations is used to solve individual equations. The equation for momentum balance (2) and subglacial drainage (4)–(9) are discretised using conservative finite differences with implicit timesteps (Katz et al., 2007). The equation for mass balance (1) is discretised with a third-order upwind scheme. We also use a linear subgrid interpolation at the grounding line (Pattyn, 2003).*

*At each timestep, we use a segregated loop to solve our model. That is, the equations for mass balance (H), momentum balance (u), and subglacial water content ($h_s$ and $h_w$) are solved in three independent steps. To ensure that the combined solution at the current timestep is converged, the steps are iterated using a Picard scheme until convergence is achieved. That is, at each timestep $t_i$ we calculate $u(t_i^1)$, $H(t_i^1, x)$, $h_s(t_i^1, x)$ and $h_w(t_i^1, x)$ in the first iteration and use these to continue to calculate $u(t_i^2, x)$, $H(t_i^2, x)$, etc until further iteration does not alter the solution at the current timestep. The size of the timestep is adjusted to achieve convergence of the individual SNES solvers as well as convergence of the segregated solution in no more than 10 Picard iterations.*

*The implemented scheme is of $O(\Delta x)$ accuracy, and comparison of steady-state results and timeseries under grid refinement are shown in figure A1. While the qualitative behaviour is consistent for values of $\Delta x \lesssim 1000$ m, convergence requires resolution at 10s of meters. Solutions shown here were calculated at resolutions of $\Delta x = 100$ m or finer. Simulation times depend on the hydraulic conductivity parameter and of course model resolution. For $\kappa \lesssim 1$ and a grid spacing of $\Delta x = 100$ m, $10^4$ model years are computed in about 24 hours on a single node. Simulations with $\kappa \gg 1$ require significantly more time (up to two weeks) due to limitations on the timestep.*

**Minor suggestions:**

- L1: semicolon unnecessary

  Changed.

- L6: is there a good reason to use "surge" and "oscillation" terminology separately here. It gives the false impression that these are strongly different phenomena.

  A comment also raised by the other reviewer. We will change surge to oscillations were appropriate.

- L12: mass discharge occurs in regions

  Changed.

- L44: here and throughout this is referred to as the "undrained bed model", but historically Tulaczyk called this the "undrained plastic bed model". Is there a reason for dropping "plastic"?

  Our model is undrained but not fully plastic (see sliding law in equation (3)). Moreover, some of the studies we cite use an undrained but not plastic model, for example Fowler and Johnson (1996).

- L45: subglacial water discharge

  Changed.

- L51: due to flow, water freezes and the ice Changed.

- Figure 1 - great figure! Thanks! :)

- L64: indicate here that this limit goes to the solution given by Tsai Added the following sentence:

  *This is the limiting case considered in Tsai et al. (2015).*

- L68: at intermediate hydraulic conductivities Changed.

- L76: This is confusing because it implies that the glacier length is a constant 1000 km, but this is merely the scale, and the grounding line evolves (as you explain below). Perhaps reword this. The sentence is now:

  *The computational domain is  of length $L_x = 1000$ km with a downward sloping bed at elevation $b = z_0 - z_1 x$.*

- L99: only a few models Changed.

- L100: subglacial water mass Changed.

- L109: could reword this to point out that this isn't actually a canal model, just a model for down-gradient porous water flow through till which could incorporate the bulk effect of canals through increased hydraulic conductivity

  Changed the sentence to be more specific:

  *Equation (5) is similar to a model of distributed water flow in a system of  subglacial conduits ("canals") eroded into soft beds (Walder and Fowler, 1994).*

- L142: sediment from freezing Changed.

- Figure 2: Its a bit confuding as to whether the two columns in the intermediate Kappa range are for different values of Kappa? Perhaps should indicate they are different ranges within the intermediate range? Columns a to d in figure 2 show the same values of $\kappa$ as figure 4. We will revise the figure to clarify this. We understand "intermediate" as between $\kappa \gg 1$ and $\kappa \ll 1$, but of course that does not mean $\kappa = 1$.

- L149: Please provide a physical justification for why it makes sense that effective pressure goes to zero at the grounding line Edited the sentence to read:

  *Assuming ice overburden pressure equal to water pressure at the grounding line, w set the effective pressure to zero at the grounding line [...]*

- L170: please indicate the time scales of the upstream traveling wave Added the speed at which upstream and downstream traveling waves propagate.

  *Notably, hydraulically controlled oscillations are characterised by a quasi-simultaneous speed-up of the entire ice stream in less than 2 yrs (figure 2d), while thermally controlled oscillations exhibit an activation wave, which travels upstream at about 4 km/yr during the speed-up phase (figure 2a). In the regime between these two limiting cases ($\kappa \sim 1$), ice stream activation can occur by downstream propagation of an activation wave at about 2 km/yr (figure 2b).*

- L172: different regimes Changed

- L184: N is indicate in row (a), not row (b)? Both. Bed is coloured according to $N$, but it is plotted also in b to emphasize the boundary layer.

- L188: combine this paragraph with the previous one? Done.

- L190: leads to a local build up in the Changed

- L204: "mechanical barrier upstream of the grounding line" you explain what you mean by this later, but it would be more useful to describe exactly what you mean by this here Changed this to

  *This is due to the region of elevated effective pressure and basal shear  upstream of the grounding line becoming less pronounced (compare the regions of elevated effective pressure in figure 4a$_4$ and b$_4$).*

- L209: a few sentences could be added here describing the variation in oscillation period in more detail - it varies from X yrs to y years over this range of kappa... Extended the sentence to

  *The period of hydraulically controlled oscillations decreases from $\approx 2000$ yrs at $\kappa = 50$ to a minimum of  $\approx 300$ yrs at $\kappa = 1$; with further reduction of $\kappa$ the oscillation period increases again up to a period of $\approx 800$ yrs at $\kappa = 10^{-2}$ (figure 5e).*

- Figure 4: not sure what the $N_c$ is doing above the colobar for the $N$ column

  $N_c$ indicated the upper limit of the colorbar. This will be changed to the numerical value.

- L226: see comment for L204 about the "buttress" which is a different term than was used before Changed to

  *As in the hydraulically controlled oscillations, the region of elevated effective pressure and basal shear stress downstream of the surge front acts as a buttress to the overall flow.*

- L249: would be prudent to also cite Fowler & Schiavi 1998 where many of these ideas originated Done.

- Figure 5b/e: I don't think the y-axis needs a log-scale, obscures some of the variation that occurs here We will adjust figures as appropriate.

- L263: yes, but you hold everything else constant, so this line reads as a bit absolute given that you don't (in this study) vary other parameters Changed "controls" to "alters".

- L270: surging mountain glaciers? Changed.

- L274: basal temperature gradient? Changed gradient to transition.

- L276: by upstream-traveling activation waves Changed to

  *Recent observations have also shown that some surges are instead characterised by  activation waves travelling upglacier*

- L282: in some sense your study shows this as well, since climatic and gemoetric factors also enter into Kappa, which is the relevant parameter of this study Indeed. Added the following sentence

  *This is supported by the appearance of the accumulation rate a and the characteristic ice sheet extent $L_x$ in the ratio of water velocity to ice velocity $\kappa$ (14a).*

- L288: do no always apply when basal Changed.

- L289: The study by Robel et al. 2016 in TC shows that thermal oscillations can temporarily mitigate a positive feedback of grounding line flux and thickness on retrograde slopes. May be useful to make the connection to MISI-style arguments here

  Added the following sentence:

  *Previous studies investigating grounding line dynamics with evolving boundary conditions have also shown that thermal oscillations can temporarily stabilise grounding lines on retrograde slopes (Robel et al., 2016).*

- L291: can change on decadal to centennial time scales (since this is the time scale for passage of activation/deactivation waves, not necessarily the full period of an oscillation) Changed.

- L293: uncertainty in how Changed.

- L294: similar point to #3 above - your model can be used to speak to the computational requirements for simulating these kinds of variability, which currently is a bit glossed over in appendix and not discussed much at all in main text. Would be useful for modelers interested in incorporating these dynamics in large-scale models to have a sense for resolution they should be aiming for. We have extended the description of the numerical solution to provide additional information.

- L333: there is an interesting literature on interactions between ice streams, particularly the water piracy hypothesis (Anadakrishnan and Alley papers 1994 and 1997) that would be worth discussing in the context of your results We have extended the sentence to account for the water piracy hypothesis

  *Another limitation of the width-averaged approach is that it precludes the formation of ice ridges, whose surface slopes affect the ice stream stability by altering the hydraulic gradient (Kyrke-Smith et al., 2014) and the interaction of neighbouring ice streams. For example, it has been hypothesised that changes in subglacial water pathways might have contributed to the shutdown of Kamb ice stream approximately 140 years ago (e.g., Anandakrishnan and Alley, 1997).*

- L347: steady-streaming Left this as is as streaming is the noun here.

- L349: at period from a few centuries to millennia Changed.

- L356: $O(\Delta x)$ accuracy Changed.

Citation: https://doi.org/10.5194/egusphere-2025-204-RC1

**References**

R. B. Alley, D. D. Blankenship, C. R. Bentley, and S. T. Rooney. Deformation of till beneath ice stream B, West Antarctica. *Nature*, 322:57–59, 1986. doi: 10.1038/322057a0.

R. B. Alley, D. Blankenship, C. Bentley, and S. Rooney. Till beneath ice stream B, 3, Till deformation: Evidence and implications. *Journal of Geophysical Research*, 92:8921–8930, doi:10.1029/JB092iB09p08921, 1987.

S. Anandakrishnan and R. B. Alley. Stagnation of ice stream C, West Antarctica by water piracy. *Geophysical Research Letters*, 24(3):265–268, doi:10.1029/96GL04016, 1997.

S. Balay, S. Abhyankar, M. F. Adams, S. Benson, J. Brown, P. Brune, K. Buschelman, E. M. Constantinescu, L. Dalcin, A. Dener, V. Eijkhout, J. Faibussowitsch, W. D. Gropp, V. Hapla, T. Isaac, P. Jolivet, D. Karpeev, D. Kaushik, M. G. Knepley, F. Kong, S. Kruger, D. A. May, L. C. McInnes, R. T. Mills, L. Mitchell, T. Munson, J. E. Roman, K. Rupp, P. Sanan, J. Sarich, B. F. Smith, S. Zampini, H. Zhang, H. Zhang, and J. Zhang. PETSc Web page. `https://petsc.org/`, 2023. URL `https://petsc.org/`.

I. Bartholomew, P. Nienow, A. Sole, D. Mair, T. Cowton, and M. A. King. Short-term variability in greenland ice sheet motion forced by time-varying meltwater drainage: Implications for the relationship between subglacial drainage system behavior and ice velocity. *Journal of Geophysical Research: Earth Surface*, 117(F3), 2012. doi: 10.1029/2011JF002220.

D. D. Blankenship, C. R. Bentley, S. T. Rooney, and R. B. Alley. Seismic measurements reveal a saturated porous layer beneath an active Antarctic ice stream. *Nature*, 322:54–57, 1986. doi: 10.1038/322054a0.

D. D. Blankenship, C. R. Bentley, S. T. Rooney, and R. B. Alley. Till beneath ice stream B. I - Properties derived from seismic travel times. II - Structure and continuity. III - Till deformation - Evidence and implications. IV - A coupled ice-till flow model. *Journal of Geophysical Research*, 92: 8903–8940, 1987. doi: 10.1029/JB092iB09p08903.

G. Catania, C. Hulbe, H. Conway, T. A. Scambos, and C. F. Raymond. Variability in the mass flux of the Ross ice streams, West Antarctica, over the last millennium. *Journal of Glaciology*, 58:741–752, 2012. doi: 10.3189/2012JoG11J219.

J. E. Christian, A. A. Robel, and G. Catania. A probabilistic framework for quantifying the role of anthropogenic climate change in marine-terminating glacier retreats. *The Cryosphere*, 16(7):2725–2743, 2022. doi: 10.5194/tc-16-2725-2022. URL `https://tc.copernicus.org/articles/16/2725/2022/`.

T. T. Creyts and C. G. Schoof. Drainage through subglacial water sheets. *Journal of Geophysical Research*, 114:F04008, doi:10.1029/2008JF001215, 2009.

C. Dow, N. Ross, H. Jeofry, K. Siu, and M. Siegert. Antarctic basal environment shaped by high-pressure flow through a subglacial river system. *Nature Geoscience*, 15(11):892–898, 2022. doi: 10.1038/s41561-022-01059-1.

H. Engelhardt and B. Kamb. Basal hydraulic system of a West Antarctic ice stream: constraints from borehole observations. *Journal of Glaciology*, 43:207–230, 1997.

H. Engelhardt, N. Humphrey, B. Kamb, and M. Fahnestock. Physical Conditions at the Base of a Fast Moving Antarctic Ice Stream. *Science*, 248:57–59, 1990. doi: 10.1126/science.248.4951.57.

G. E. Flowers. Modelling water flow under glaciers and ice sheets. *Proceedings of the Royal Society of London A: Mathematical, Physical and Engineering Sciences*, 471(2176), 2015. ISSN 1364-5021. doi: 10.1098/rspa.2014.0907.

A. C. Fowler and C. Johnson. Ice-sheet surging and ice-stream formation. *Annals of Glaciology*, 23: 68–73, 1996.

A. O. Hager, M. J. Hoffman, S. F. Price, and D. M. Schroeder. Persistent, extensive channelized drainage modeled beneath Thwaites Glacier, West Antarctica. *The Cryosphere*, 16(9):3575–3599, 2022. doi: 10.5194/tc-16-3575-2022.

I. J. Hewitt. Modelling distributed and channelized subglacial drainage: the spacing of channels. *Journal of Glaciology*, 57(202):302–314, 2011. doi: 10.3189/002214311796405951.

C. Hulbe and M. Fahnestock. Century-scale discharge stagnation and reactivation of the Ross ice streams, West Antarctica. *Journal of Geophysical Research*, 112:F03S27, 2007. doi: 10.1029/2006JF000603.

R. Katz, M. Knepley, B. Smith, M. Spiegelman, and E. Coon. Numerical simulation of geodynamic processes with the portable extensible toolkit for scientific computation. *Physics of the Earth and Planetary Interiors*, 163(1):52–68, 2007. ISSN 0031-9201. doi: 10.1016/j.pepi.2007.04.016. Computational Challenges in the Earth Sciences.

T. M. Kyrke-Smith, R. F. Katz, and A. C. Fowler. Subglacial hydrology and the formation of ice streams. *Proceedings of the Royal Society A: Mathematical, Physical and Engineering Science*, 470 (2161):20130494, 2014. doi: 10.1098/rspa.2013.0494.

T. W. Lambe and R. V. Whitman. *Soil mechanics*, volume 10. John Wiley & Sons, 1991. ISBN 978-0-471-51192-2.

J. R. Leeman, R. D. Valdez, R. B. Alley, S. Anandakrishnan, and D. M. Saffer. Mechanical and hydrologic properties of Whillans Ice Stream till: Implications for basal strength and stick-slip failure. *Journal of Geophysical Research: Earth Surface*, 121(7):1295–1309, 2016. doi: 10.1002/2016JF003863.

D. R. MacAyeal. Large-scale ice flow over a viscous basal sediment - Theory and application to ice stream B, Antarctica. *Journal of Geophysical Research*, 94:4071–4087, 1989. doi: 10.1029/JB094iB04p04071.

E. Mantelli, M. B. Bertagni, and L. Ridolfi. Stochastic ice stream dynamics. *Proceedings of the National Academy of Sciences*, 113(32):E4594–E4600, 2016. doi: 10.1073/pnas.160036211.

F. S. L. Ng. Coupled ice–till deformation near subglacial channels and cavities. *Journal of Glaciology*, 46(155):580–598, 2000. doi: 10.3189/172756500781832756.

F. Pattyn. A new three-dimensional higher-order thermomechanical ice sheet model: Basic sensitivity, ice stream development, and ice flow across subglacial lakes. *Journal of Geophysical Research*, 108: 2382, doi:10.1029/2002JB002329, 2003.

L. E. Peters, S. Anandakrishnan, R. B. Alley, J. P. Winberry, D. E. Voigt, A. M. Smith, and D. L. Morse. Subglacial sediments as a control on the onset and location of two Siple Coast ice streams, West Antarctica. *Journal of Geophysical Research*, 111:B01302, doi:10.1029/2005JB003766, Jan. 2006. doi: 10.1029/2005JB003766.

R. Retzlaff and C. R. Bentley. Timing of stagnation of Ice Stream C, West Antarctica, from short-pulse radar studies of buried surface crevasses. *Journal of Glaciology*, 39(133):553–561, 1993. doi: 10.3189/S0022143000016440.

A. Robel, C. Schoof, and E. Tziperman. Rapid grounding line migration induced by internal ice stream variability. *Journal of Geophysical Research*, 119(11):2430–2447, doi:10.1002/2014JF003251, 2014.

A. A. Robel, E. DeGiuli, C. Schoof, and E. Tziperman. Dynamics of ice stream temporal variability: Modes, scales, and hysteresis. *Journal of Geophysical Research*, 118(2):925–936, 2013. doi: 10.1002/jgrf.20072.

A. A. Robel, C. Schoof, and E. Tziperman. Persistence and variability of ice-stream grounding lines on retrograde bed slopes. *The Cryosphere*, 10(4):1883–1896, 2016. doi: 10.5194/tc-10-1883-2016. URL https://tc.copernicus.org/articles/10/1883/2016/.

A. A. Robel, G. H. Roe, and M. Haseloff. Response of marine-terminating glaciers to forcing: Time scales, sensitivities, instabilities, and stochastic dynamics. *Journal of Geophysical Research: Earth Surface*, 123(9):2205–2227, 2018. doi: 10.1029/2018JF004709.

H. Röthlisberger. Water pressure in subglacial channels. *Journal of Glaciology*, 11(62):177–203, 1972. doi: 10.3189/S0022143000022188.

C. Schoof. Ice-sheet acceleration driven by melt supply variability. *Nature*, 468(7325):803–806, 2010. doi: 10.1038/nature09618.

D. M. Schroeder, D. D. Blankenship, D. A. Young, A. E. Witus, and J. B. Anderson. Airborne radar sounding evidence for deformable sediments and outcropping bedrock beneath Thwaites Glacier, West Antarctica. *Geophysical Research Letters*, 41(20):7200–7208, 2014. doi: 10.1002/2014GL061645.

O. Sergienko and M. Haseloff. 'Stable' and 'unstable' are not useful descriptions of marine ice sheets in the Earth's climate system. *Journal of Glaciology*, page 1–17, 2023. doi: 10.1017/jog.2023.40.

V. C. Tsai, A. L. Stewart, and A. F. Thompson. Marine ice-sheet profiles and stability under Coulomb basal conditions. *Journal of Glaciology*, 61(226):205–215, 2015.

J. S. Walder and A. Fowler. Channelized subglacial drainage over a deformable bed. *Journal of Glaciology*, 40(134):3–15, 1994. doi: 10.3189/S0022143000003750.

K. Warburton, D. Hewitt, and J. Neufeld. Tidal grounding-line migration modulated by subglacial hydrology. *Geophysical Research Letters*, 47(17):e2020GL089088, 2020. doi: 10.1029/2020GL089088.

---

## Author Comment (AC2)

**Response to reviewer 2 (Elise Kazmierczak)**

M. Haseloff, I. J. Hewitt, R. F. Katz

May 19, 2025

We thank the reviewer for their thoughtful and constructive review. In this document, reviewer text is in black and our response to it is in blue. Manuscript text is in italics, with original text in black and changes in red.

The paper is well-written and demonstrates a high level of scientific quality. The topic is of significant interest and aligns well with current research on the influence of subglacial hydrology on ice motion, particularly within the context of the Antarctic ice sheet. This area of study has gained renewed attention in recent years, addressing gaps in the existing literature.

The research demonstrates the impact of subglacial hydrology on marine ice stream dynamics, coupling various systems in a positive feedback loop involving ice flow, basal heat dissipation, and basal lubrication. By using a simple flow-line model, the authors explore different subglacial phenomena arising from the presence of subglacial water in a soft bed system. Their results emphasize the dependence of ice stream dynamics on basal conditions.

I recommend the publication of this paper following a few minor revisions.

I have structured the comments into general comments and line-by-line comments. The general comments are suggestions to improve clarity for a new reader, curiosity questions, or suggestions of improvements that are recurrent throughout the paper.

**1 General comments and suggestions**

- I suggest to summarize in a simple and clear way the model at the beginning of the paper. Indeed, it is a bit complicated during the first read of the paper to understand well in terms of subglacial hydrology what is considered and what is not. For example, please specify that it's a model only applied on a soft bed and which encompasses inefficient and efficient subglacial drainage systems through a hydraulic conductivity factor that increases with decrease of the effective pressure. No switch between efficient and inefficient drainages is assumed, as well as no channels at higher subglacial water flux. Furthermore, no interaction between subglacial water and ocean water and no vertical infiltration are considered. I also think you could more explicitly summarize what subglacial processes the kappa factor encompass.

In the introduction, we have expanded the paragraph describing our model as follows:

*The goal of this study is to investigate the simplest possible feedback between fast flow, heat dissipation, and basal lubrication, and its role in marine ice sheet dynamics. We use a model that includes the positive feedback between fast flow and basal lubrication through both storage of water in the subglacial sediment as well as evacuation of water through an active subglacial drainage system.* *Subglacial water flow is modelled with a Darcy-style flux law. The hydraulic conductivity is the quotient of a conductivity factor and the effective pressure, i.e., it is assumed to increase as the effective pressure decreases, as is common for distributed systems. The overall* *balance of* *storage and subglacial discharge*  *is determined by the hydraulic conductivity* *factor* . *At very low hydraulic conductivity, the model reproduces the undrained bed model, and water content of the bed is determined by the local energy balance. At very high hydraulic conductivities, hydraulic gradients cannot develop at the bed; in this limit the effective pressure at the bed is set by the ice and bed geometry alone, independent of the local melt rate.*

In the discussion, we added to section 4.4 on model limitation:

*We also ignore the role of ice shelves here. Buttressing ice shelves can alter marine ice stream dynamics (Gudmundsson et al., 2012; Gudmundsson, 2013; Haseloff and Sergienko, 2018, 2022). In addition, at the grounding line, subglacial hydrology can interact with the ocean in multiple ways not taken into account here, for example through seawater infiltration and the initialisation of subshelf meltwater plumes which might alter ice shelf buttressing (e.g., Jenkins, 2011; Robel et al., 2022; Ehrenfeucht et al., 2024).*

We have also changed section 4.3 to more explicitly state that we do not consider channelisation:

*For high water discharge conditions, it is plausible that a transition from distributed to channelised drainage might occur (Walder and Fowler, 1994; Röthlisberger, 1972; Hager et al., 2022; Dow et al., 2022). Under these conditions, we expect the net discharge to increase with effective pressure, rather than decrease.  In Greenland, where abundant meltwater can drain to the bed, increasing channelisation leads to a strengthening of the bed at the end of the melt season (Schoof, 2010; Bartholomew et al., 2012). We  also expect the conductivity to vary locally, so that we interpret $K_{d,eff}$ as an effective hydraulic conductivity, averaged over spatial scales resolvable in ice sheet models.*

- Also, I do not understand whether or not $h_w$ & $h_s$ are limited. That is the case, please mention it.

  The water content $h_w$ can freely evolve, but values below 0 are unphysical; once the basal water content is zero, we need to solve for basal temperature, rather than water content. However, this scenario does not occur with our parameter choices. The thickness of the hydraulically-active, drained sediment layer $h_s$ is bounded between 0 (fully frozen) and $h_0$ (completely unfrozen). To clarify this, we have made the following change:

  *Note that the thickness of the unexpanded, hydraulically active sediment layer $h_s$ is bounded between 0 (completely frozen) and $h_0$ (completely unfrozen). our  parameter choices prevent the  basal sediment from freezing entirely ($h_w =h_s = 0$).*

- Some values used in this research (e.g., constant coefficients in Table 1 or values of kappa used in experiments) are not referenced, and there is no explanation regarding the choice of their values. Please provide the references where applicable and add comments about the choices that had to be made.

  We added the following information to the table caption:

  *Geometric ($W$, $z_0$, $z_1$), environmental ($a$, $T_s$, $A$, $q_{geo}$), and basal parameters ($h_0$, $C$) are chosen to be representative of Siple Coast ice streams and for comparability with previous studies (e.g., Robel et al., 2013, 2014; Tsai et al., 2015). Subglacial drainage parameters ($e_r$, $e_c$, $N_r$) are based on Tulaczyk et al. (2000).*

- I think it's important to clearly specify that the processes you talk about are subglacial processes. Therefore, make sure to add the term "basal" with "melt" and "subglacial" with "water" when necessary. The same goes for the term "discharge" — make sure to specify whether it refers to water or ice.

  Done.

- In general, in the presentation of the results, I suggest emphasizing the quantitative values of the mentioned variations and referencing the relevant figures/videos more regularly.

  The manuscript has been edited for more consistent referencing and quantitative description, where appropriate .

- Does one of your limit cases could be interpreted as that of a hard bed system?

  While we focus on soft-bedded conditions, aspects of the results shown here extend to hard-bedded systems: Regularized Coulomb friction laws have been proposed for hard beds as well

(Helanow et al., 2020), and in the limit of high effective pressure, the basal shear stress reproduces a Weertman sliding law (Weertman, 1957). The main difference between hard- and soft bedded system might lie in the details of the subglacial drainage system, but if the flow is essentially Darcian with the hydraulic conductivity increasing with decreasing effective pressure, then qualitatively similar dynamics should be expected as long as no channelisation occurs. In this case melt would lead to an overall strengthening off the bed. In the manuscript, we acknowledge this in the discussion:

*For high water discharge conditions, it is plausible that a transition from distributed to channelised drainage might occur (Walder and Fowler, 1994; Röthlisberger, 1972). Under these conditions, we expect the net discharge to increases with effective pressure, rather than decreases.  In Greenland, where abundant surface meltwater can drain to the bed, increasing channelisation leads to a strengthening of the bed at the end of the melt season (Schoof, 2010; Bartholomew et al., 2012).*

- Have other geometries (slope/shape) of beds been investigated?

  Not systematically, as the goal of this study was to use the simplest possible model. Other studies have emphasized the importance of basal topography on marine ice stream dynamics (e.g., Sergienko and Wingham, 2021, 2024). Similarly, the use of a width-integrated model prevents the formation of lateral hydraulic gradients, which might be important for the observed patterning of the Siple Coast ice streams (Kyrke-Smith et al., 2014). Extension of this model to other geometries is a potential avenue for future work. We acknowledge these limitations in the discussion, where we write:

  *Another limitation of the width-averaged approach is that it precludes the formation of ice ridges, whose surface slopes affect the ice stream stability by altering the hydraulic gradient (Kyrke-Smith et al., 2014).*

- In terms of results, more details could be provided regarding the timing of the experiments. For example, I don't understand why, in Figure 2, there is a 800-year difference between d1 and d2, while in the other figures, the results are shown to take place over a scale of a few years. I also think a comment on the initial and final conditions would be interesting.

  The profiles in figure 2 were mainly intended for illustrative purposes, as they show the same information as figures 3 and 4. Times are relative to an initial point $t_0$ within the oscillation cycle shown in figures 4 (a: $t_0 = 6675$ yrs, b: $t_0 = 6300$ yrs, c: $t_0 = 6200$ yrs, d: $t_0 = 5800$ yrs); the displayed profiles are selected to illustrate the qualitative differences in ice stream behaviour. We will change the times to match the information in figure 4.

- What is the computation time for these models?

  Simulation times depend on the hydraulic conductivity parameter and of course model resolution. For $\kappa \lesssim 1$, and a grid spacing of $\Delta x = 100$ m, $10^4$ model years are computed in about 24 hours on a single node. Simulations with $\kappa \gg 1$ require significantly more time (up to two weeks) due to limitations on the timestep. This information has been added to the description of the numerical model.

- In your text, you explain well that kappa is an average used to represent in 1D hydrological systems that would be found in 2D. You also mention that multiple systems can exist for $N$. However, depending on the drainage system, for the same flow, a different effective pressure could be obtained (cf. Fig. 4 Walder and Fowler, 1994). Maybe you discuss how your model could include other drainage systems.

  Figure 4 in Walder and Fowler (1994) shows effective pressure vs subglacial discharge of a mountain glacier-like scenario (large surface slope of $\sin \alpha = 0.1$) and an ice sheet-like scenario (small surface slope of $\sin \alpha = 0.001$), with a simplified version of the latter adopted in our study. While

we are interested in the simplest possible model that can illustrate the role of subglacial hydrology and be used to identify the leading order controls (represented through $\kappa$) extension of our model to more sophisticated drainage models is an exciting research direction. For such an extension, explicitly resolving the across-flow direction becomes crucial and continuum-mechanical models capable of including such dynamics are beginning to emerge (e.g., Bueler and van Pelt, 2015; Sommers et al., 2018).

**2  Line-by-line comments**

**2.1  Introduction**

- L6: I don't understand why a hydraulically controlled motion is called a "surge", while a thermally controlled motion is referred to as an "oscillation". Shouldn't we use the same term, or are these two different phenomena?

  A comment also raised by the other reviewer. We will change surge to oscillations were appropriate.

- L25: Maybe add something that explains why existing observational work does not address the interplay between subglacial hydrology and marine ice-sheet dynamics, such as 'because of the lack of direct observation.'

  Done.

- L27: Maybe add: Gregov, T., Pattyn, F., & Arnst, M. (2023). Grounding-line flux conditions for marine ice-sheet systems under effective-pressure-dependent and hybrid friction laws. Journal of Fluid Mechanics, 975, A6.

  Done.

- L40: By reading this sentence, one might think that in an extreme case, this feedback loop never stops. Is it possible to add something to moderate it, like "up to a certain point"?

  We changed the sentence to:

  *The dominant positive feedback mechanism then involves melting at the base of the ice; faster sliding leads to more heat dissipation, which in turn produces additional melt water that reduces basal drag and permits even faster sliding until other processes suppress further weakening of the bed.*

- L44: add something like "Which is often the case because of the low porosity of this material"

  Done.

- L52: "subglacial water" –"and water content of the bed composed by till"

  We added the first part but we are opting to leave the second part of the sentence as is as the cited studies are not specific to till beds.

- L63: Give a numerical value for the low hydraulic conductivity

  Done: added ($K \ll 10^{-3}$ m s$^{-1}$)

- L64: At what value of hydraulic conductivity is the limiting case obtained?

  Added ($K \gg 10^{-3}$ m s$^{-1}$)

- L68: finite and intermediate hydraulic conductivities

  Changed finite to intermediate as suggested also by Reviewer 1.

**2.2 The Model**

- **L76:** It's not clear that the length refers to the domain and not to the flowline.

  Changed the sentence as follows:

  *The computational domain is  of length $L_x = 1000$ km with a downward sloping bed at elevation $b = z_0 - z_1 x$.*

- **L77:** depth or thickness to clarify the word "content"

  Changed the sentence as follows:

  *We solve for velocity $u$, ice thickness $H$, water content of the bed $h_w$ (given as water column thickness), and thickness of unexpanded, hydraulically active till layer $h_s$, which can evolve due to freezing and melting in the sediment, see figure 1.*

- **L86-87:** For $n$, keep "rheological" as used in the text or "viscosity" as in the table, but not both—avoid using two different terms. The value of epsilon is missing in the table. What is the value, unit, and source of epsilon? Also, please provide more details about Cw.

  We changed the description of $n$ in the table and added the entry of $\varepsilon$. Regularisation of the viscosity term to avoid the viscosity to become infinite is common practice in ice sheet modelling (e.g., Schoof, 2006; Bueler and Brown, 2009). We have added the references. $C_w$ arises in the width-integration of the ice flow equations. We have added a reference to Hindmarsh (2012) for further details.

- **L88:** Refer to some of these recent studies.

  Note that Zoet and Iverson (2020) is already referenced in the sentence. We have added Helanow et al. (2021) as reference for a hard bed.

- **L92:** Add a note that explains that since $\mu$ is constant at high effective pressure, you obtain a Weertman-style sliding law. Also, include a reference explaining why m = 1/3 is used.

  Note that we already write:
  *For $\mu N \gg C|u|^m$ this reproduces a Weertman-style sliding law $\tau_b \sim C|u|^m$ (Weertman, 1957), [...]*

  Because the exponent in the Weertman sliding law is related to the creep of ice, it is often set to $m = 1/n$ (Brondex et al., 2017; Weertman, 1974; Schoof, 2007b; Gudmundsson et al., 2012; Pattyn et al., 2012, 2013). We changed the sentence to:

  *We use a regularized version of the slip law used in Tsai et al. (2015),*

  $$\tau_b = \frac{C|u|^m \mu N}{C|u|^m + \mu N} \frac{u}{|u|}, \tag{1}$$

  *with $m = 1/n$ (e.g., Weertman, 1974; Pattyn et al., 2012; Brondex et al., 2017).*

- **L99:** Maybe add the reference: Shreve, R. L. (1972). Movement of water in glaciers. Journal of Glaciology, 11(62), 205-214.

  Done.

- **L100:** Maybe add references like van der Wel et al., 2013, Bougamont et al., 2014 and Bueler and van Pelt, 2015.

  van der Wel, N., Christoffersen, P., & Bougamont, M. (2013). The influence of subglacial hydrology on the flow of Kamb Ice Stream, West Antarctica. Journal of Geophysical Research: Earth Surface, 118(1), 97-110.

  Bougamont, M., Christoffersen, P., Hubbard, A. L., Fitzpatrick, A. A., Doyle, S. H., & Carter, S. P. (2014). Sensitive response of the Greenland Ice Sheet to surface melt drainage over a soft bed. Nature communications, 5(1), 5052.

Bueler, E., & van Pelt, W. (2015). Mass-conserving subglacial hydrology in the Parallel Ice Sheet Model version 0.6. Geoscientific Model Development, 8(6), 1613-1635.

Done.

- L100-L102: subglacial water

  Done.

- L103: If I understand correctly, the properties of the bed itself are not modified. I would suggest modifying the sentence to: "The conductivity only depends on the water content [...] and the properties of the bed are kept constant".

  Arguably, the effective pressure is a property of the bed, which changes in the model, as does the void space, which depends on the effective pressure. We have changed the sentence subtly thus:

  *We assume that the hydraulic conductivity only depends on the water content of the bed, as is common for distributed systems (Hewitt, 2011).*

- L109: The difference of pressures allows canals to deform the soft bed. So I will modify the sentence by "eroded and deformed".

  We have opted to leave the sentence as is, as the current formulation does not exclude the possibility of canal deformation, but avoids discussion of the contentious topic how much till beneath conduits actually deforms (see for example Damsgaard et al., 2017).

- L120: For the values of the model-specific constants, I suggest adding the sources of the values used.

  The following information has been added to the table caption:

  *Geometric ($W$, $z_0$, $z_1$), environmental ($a$, $T_s$, $A$, $q_{geo}$), and basal parameters ($h_0$, $C$) are chosen to be representative of Siple Coast ice streams and for comparability with previous studies (e.g., Robel et al., 2013, 2014; Tsai et al., 2015). Subglacial drainage parameters ($e_r$, $e_c$, $N_r$) are based on Tulaczyk et al. (2000).*

- L140: Add a reference for this.

  This is to be understood as a possible (logical) consequence of the model formulation, not a comment on observations. We have changed the sentence to clarify:

  * It is conceivable that freezing could occur at smaller void ratios if subglacial drainage has removed water before the onset of freezing so that $N > N_c$.*

- L14x: Add numbers/letters in Eq. 10 conditions for more clarity. I propose 10b.

  Since Eq. 10 is one equation, we reference the subcases with subscripts, e.g., $(10)_1$ etc. We have added explicit reference to the subcases to make the paragraph clearer.

- L153: Maybe add that the GL and the calving front are the same (even though it is mentioned on line 336).

  We have added the following sentence immediately after equation 12:

  *In the numerical model, we assume that an unbuttressed ice shelf fills the domain from the grounding line to the domain boundary, where we apply $(12)_2$. This is mathematically equivalent to (12), which assumes that the ice sheet terminates at the grounding line (e.g., Schoof, 2007a).*

**2.3 Results**

- L174: basal melt water

  Changed.

- L 175: In the limiting case of quasi-infinite conductivity of the hydraulic system, I do not understand how water can be evacuated without hydraulic gradients. From my understanding, this limit case corresponds to the height above buoyancy model. Could you provide me with further explanations?

Indeed, the height above buoyancy model corresponds to assuming a quasi-infinite hydraulic permeability of the bed. Assume $\kappa \to \infty$, then equation (15) states:

$$\frac{\partial \Phi}{\partial x} = -\frac{\mu N}{\kappa a L_x h_s} q_w \to 0 \quad \text{for} \quad \kappa \to \infty.$$

However, this does not define the flux, which is now given through equation (4):

$$\frac{\partial h_w}{\partial t} + \frac{\partial q_w}{\partial x} = m_b$$

In steady state, this gives

$$q_x(x) = \int_0^x m_b \ \mathrm{d}x$$

Maybe more intuitive is to think of this as $\kappa \gg 1$. For large $\kappa$, infinitely small gradients in the hydraulic gradient are sufficient to drive a large water flux. We have changed the corresponding paragraph to be clearer:

*Consequently, $\Phi = \rho_w g b - N + \rho_i g H =$ constant. At the grounding line, boundary conditions (12) require $\Phi(x_g) = 0$. This requires the effective pressure to adjust such that $N = \rho_i g H + \rho_w g b$, that is, the subglacial water content is set by the ice and bed geometry alone, and is independent of the basal melt rate (which is positive throughout, see figure $2_1$).* *This limit is also referred to as height-above-buoyancy or height-above-floatation model (Van der Veen, 1987; Asay-Davis et al., 2016; Kazmierczak et al., 2022) and it.* *This leads to high effective pressure and low water content throughout most of the domain (figure $3b_1$) apart from a small region near the grounding line. Stable steady-state solutions exist, characterised by high ice thickness and surface slopes (figure $3a_1$).* *In steady state, the water flux is given by (4), i.e., $q_w = \int_0^x m_b \ dx$.*

- L 185: you mention fig 5 before fig 4 in the text

We now reference figure 4 first in the caption of figure 2, which should justify the current ordering of figures.

- L 199: please refer to the corresponding figure to find directly the order of magnitude of variations and the time scale

Done.

- L209: Figure 5e

Changed.

- L222: subglacial water fluxes

Changed.

- L232-238: basal melt rate

Changed.

- L238-235: Please add numerical values to your analysis.

The conductive cooling term and basal melt rate are spatially and temporally variable, so that a single number cannot be assigned to these. Moreover, the aim of our study is to investigate general processes, rather than a specific glacier. Focussing on specific numbers could be misleading, as qualitatively similar results could be easily obtained with different parameter choices. That said, we now specify that the oscillation period is $\approx 800$ yrs.

- L250-256: Also, remind us which figure we should observe.

  Added references to the relevant figures and panels throughout.

- L250: "significant" – Maybe provide a numerical value, if one is available?

  rephrased to:

  *At high effective stress basal melt rates in this boundary layer dissipates significant heat are high ($\approx 10~mm~yr^{-1}$), quickly lowering its own the effective pressure, which then speeds up the flow immediately upstream.*

**2.4   Discussion**

- L261: I would nuance this statement by adding "namely"

  Unsure which statement this refers to? The sentence in question is

  *Subglacial conditions affect marine ice sheet dynamics through the basal shear stress, which depends on the effective pressure.*

  We've opted to leave it unchanged.

- L263: specify the kind of subglacial drainage system

  Changed to: *We find that the efficiency of a distributed subglacial drainage system alters controls the mode of grounding line dynamics.*

- L264: maybe remind the reader that low hydraulic conductivities lead to high subglacial water storage and reduce effective pressure

  Changed to *Lower hydraulic conductivities result in more water storage, lower effective pressure, and consequently lower basal resistance and faster flow, [...]*

- L288: indicate

  Changed.

- L299: Do you obtain such a result also because you considered more things in your "hydraulic conductivity parameter" ?

  We have completely revised this section in response to a comment by reviewer 1. The limitations and implications of the hydraulic model should now be clearer.

- L306: maybe add reference like Schroeder, D. M., Blankenship, D. D., Young, D. A., Witus, A. E., & Anderson, J. B. (2014). Airborne radar sounding evidence for deformable sediments and outcropping bedrock beneath Thwaites Glacier, West Antarctica. Geophysical Research Letters, 41(20), 7200-7208.

  Done.

- L307: and deforming the bed composed of till/sediments

  As above, we have opted to leave the sentence as is, as the current formulation does not exclude the possibility of till deformation, but avoids discussion how much the till around conduits actually deforms (see for example Damsgaard et al., 2017).

- L313: Papers like Hager 2022 and Dow 2022 consider channels in WAIS. Maybe add a sentence assuming that their existence is plausible in WAIS too.
  Hager, A. O., Hoffman, M. J., Price, S. F., & Schroeder, D. M. (2022). Persistent, extensive channelized drainage modeled beneath Thwaites Glacier, West Antarctica. The Cryosphere, 16(9), 3575-3599.
  Dow, C. F. (2022). Hidden rivers under Antarctica impact ice flow and stability. Nature Geoscience, 15, 869-870.

  These references have been added.

**2.5  Appendix**

L395: Inefficient rather than ineffective (by opposition to "efficient" in line 393).
   Changed.

**2.6  Figures and table**

General comments for the figures : Great figures! However, I always find it clearer when the extreme values of the colorbar are also indicated. Verify that all panels are mentioned and explained in the short text linked to each figure.
   Thank you! The colorbars of the figures will be updated and referencing checked.

- Figure 1: not clear that the bed is "downwards"

   We will try to make the downwards slope clearer.

- Table 1: If you don't explain all the parameters in the text, refer to the table. $\varepsilon$, $h_m$, $N_0$ are missing from it.

   We have added $\varepsilon$ and $N_0$ (which is also defined in equation (9)). However, there is no variable or parameter $h_m$, and a file search did not find it – maybe a misread?

- Figure 2: Explain column c and e.

   Done.

- Figure 3: Why showing specifically these values of kappa ? Maybe provide a rationale for that choice.

   All solutions we display were chosen for illustrative purposes. In figure (3), $\kappa = 10^5$ illustrates that the hydraulic potential is constant (red line), $\kappa = 10^2$ is the smallest value of $\kappa$ for which we still found a steady state, and $\kappa = 5 \times 10^3$ was chosen as a good intermediate example with an hydraulic potential between column a and c. We have added the following sentences to the figure caption:

   *Note that the hydraulic potential is constant for $\kappa = 10^5$ (red line in $a_1$) and gradually increases for decreasing $\kappa$ (panels $a_2$ and $a_3$). $\kappa = 10^2$ is the smallest value of $\kappa$ for which steady states exist.*

- Figure 4: basal melt rate

   Changed.

- Figure 5: Ice discharge / subglacial water

   Changed.

**References**

X. S. Asay-Davis, S. L. Cornford, G. Durand, B. K. Galton-Fenzi, R. M. Gladstone, G. H. Gudmundsson, T. Hattermann, D. M. Holland, D. Holland, P. R. Holland, D. F. Martin, P. Mathiot, F. Pattyn, and H. Seroussi. Experimental design for three interrelated marine ice sheet and ocean model intercomparison projects: Mismip v. 3 (mismip+), isomip v. 2 (isomip+) and misomip v. 1 (misomip1). *Geoscientific Model Development*, 9(7):2471–2497, 2016. doi: 10.5194/gmd-9-2471-2016. URL `https://www.geosci-model-dev.net/9/2471/2016/`.

I. Bartholomew, P. Nienow, A. Sole, D. Mair, T. Cowton, and M. A. King. Short-term variability in greenland ice sheet motion forced by time-varying meltwater drainage: Implications for the relationship between subglacial drainage system behavior and ice velocity. *Journal of Geophysical Research: Earth Surface*, 117(F3), 2012. doi: 10.1029/2011JF002220.

J. Brondex, O. Gagliardini, F. Gillet-Chaulet, and G. Durand. Sensitivity of grounding line dynamics to the choice of the friction law. *Journal of Glaciology*, 63(241):854–866, 2017. doi: 10.1017/jog.2017.51.

E. Bueler and J. Brown. The shallow shelf approximation as a sliding law in a thermomechanically coupled ice sheet model. *Journal of Geophysical Research*, 114:F03008, doi:10.1029/2008JF001179, 2009.

E. Bueler and W. van Pelt. Mass-conserving subglacial hydrology in the Parallel Ice Sheet Model version 0.6. *Geoscientific Model Development*, 8(6):1613–1635, 2015. doi: 10.5194/gmd-8-1613-2015.

A. Damsgaard, J. Suckale, J. A. Piotrowski, M. Houssais, M. R. Siegfried, and H. A. Fricker. Sediment behavior controls equilibrium width of subglacial channels. *Journal of Glaciology*, 63(242):1034–1048, 2017. doi: 10.1017/jog.2017.71.

C. Dow, N. Ross, H. Jeofry, K. Siu, and M. Siegert. Antarctic basal environment shaped by high-pressure flow through a subglacial river system. *Nature Geoscience*, 15(11):892–898, 2022. doi: 10.1038/s41561-022-01059-1.

S. Ehrenfeucht, E. Rignot, and M. Morlighem. Seawater intrusion in the observed grounding zone of petermann glacier causes extensive retreat. *Geophysical Research Letters*, 51(12):e2023GL107571, 2024. doi: 10.1029/2023GL107571. e2023GL107571 2023GL107571.

G. H. Gudmundsson. Ice-shelf buttressing and the stability of marine ice sheets. *The Cryosphere*, 7: 647–655, 2013. doi: 10.5194/tc-7-647-2013.

G. H. Gudmundsson, J. Krug, G. Durand, L. Favier, and O. Gagliardini. The stability of grounding lines on retrograde slopes. *The Cryosphere*, 6:1497–1505, 2012. doi: 10.5194/tc-6-1497-2012.

A. O. Hager, M. J. Hoffman, S. F. Price, and D. M. Schroeder. Persistent, extensive channelized drainage modeled beneath Thwaites Glacier, West Antarctica. *The Cryosphere*, 16(9):3575–3599, 2022. doi: 10.5194/tc-16-3575-2022.

M. Haseloff and O. V. Sergienko. The effect of buttressing on grounding line dynamics. *Journal of Glaciology*, 64(245):417–431, 2018. doi: 10.1017/jog.2018.30.

M. Haseloff and O. V. Sergienko. Effects of calving and submarine melting on steady states and stability of buttressed marine ice sheets. *Journal of Glaciology*, 68(272):1149–1166, 2022. doi: 10.1017/jog.2022.29.

C. Helanow, N. R. Iverson, L. K. Zoet, and O. Gagliardini. Sliding Relations for Glacier Slip With Cavities Over Three-Dimensional Beds. *Geophysical Research Letters*, 47(3):e2019GL084924, 2020. doi: 10.1029/2019GL084924.

C. Helanow, N. R. Iverson, J. B. Woodard, and L. K. Zoet. A slip law for hard-bedded glaciers derived from observed bed topography. *Science Advances*, 7(20):eabe7798, 2021. doi: 10.1126/sciadv.abe7798. URL https://www.science.org/doi/abs/10.1126/sciadv.abe7798.

I. J. Hewitt. Modelling distributed and channelized subglacial drainage: the spacing of channels. *Journal of Glaciology*, 57(202):302–314, 2011. doi: 10.3189/002214311796405951.

R. C. A. Hindmarsh. An observationally validated theory of viscous flow dynamics at the ice-shelf calving front. *Journal of Glaciology*, 58(208):375–387, 2012. doi: 10.3189/2012JoG11J206.

A. Jenkins. Convection-driven melting near the grounding lines of ice shelves and tidewater glaciers. *Journal of Physical Oceanography*, 41(12):2279–2294, 2011.

E. Kazmierczak, S. Sun, V. Coulon, and F. Pattyn. Subglacial hydrology modulates basal sliding response of the Antarctic ice sheet to climate forcing. *The Cryosphere*, 16(10):4537–4552, 2022. doi: 10.5194/tc-16-4537-2022. URL `https://tc.copernicus.org/articles/16/4537/2022/`.

T. M. Kyrke-Smith, R. F. Katz, and A. C. Fowler. Subglacial hydrology and the formation of ice streams. *Proceedings of the Royal Society A: Mathematical, Physical and Engineering Science*, 470 (2161):20130494, 2014. doi: 10.1098/rspa.2013.0494.

F. Pattyn, C. Schoof, L. Perichon, R. C. A. Hindmarsh, E. Bueler, B. de Fleurian, G. Durand, O. Gagliardini, R. Gladstone, D. Goldberg, G. H. Gudmundsson, P. Huybrechts, V. Lee, F. M. Nick, A. J. Payne, D. Pollard, O. Rybak, F. Saito, and A. Vieli. Results of the Marine Ice Sheet Model Intercomparison Project, MISMIP. *The Cryosphere*, 6(3):573–588, 2012. doi: 10.5194/tc-6-573-2012. URL `http://www.the-cryosphere.net/6/573/2012/`.

F. Pattyn, L. Perichon, G. Durand, L. Favier, O. Gagliardini, R. C. A. Hindmarsh, T. Zwinger, T. Albrecht, S. Cornford, D. Docquier, et al. Grounding-line migration in plan-view marine ice-sheet models: results of the ice2sea MISMIP3d intercomparison. *Journal of Glaciology*, 59(215): 410–422, 2013.

A. Robel, C. Schoof, and E. Tziperman. Rapid grounding line migration induced by internal ice stream variability. *Journal of Geophysical Research*, 119(11):2430–2447, doi:10.1002/2014JF003251, 2014.

A. A. Robel, E. DeGiuli, C. Schoof, and E. Tziperman. Dynamics of ice stream temporal variability: Modes, scales, and hysteresis. *Journal of Geophysical Research*, 118(2):925–936, 2013. doi: 10.1002/jgrf.20072.

A. A. Robel, E. Wilson, and H. Seroussi. Layered seawater intrusion and melt under grounded ice. *The Cryosphere*, 16:451–469, 2022. doi: 10.5194/tc-16-451-2022.

H. Röthlisberger. Water pressure in subglacial channels. *Journal of Glaciology*, 11(62):177–203, 1972. doi: 10.3189/S0022143000022188.

C. Schoof. A variational approach to ice stream flow. *Journal of Fluid Mechanics*, 556:227–251, 2006. doi: 10.1017/S0022112006009591.

C. Schoof. Marine ice-sheet dynamics. Part 1. The case of rapid sliding. *Journal of Fluid Mechanics*, 573:27–55, 2007a. doi: 10.1017/S0022112006003570.

C. Schoof. Ice sheet grounding line dynamics: Steady states, stability, and hysteresis. *Journal of Geophysical Research*, 112(F3), 2007b. doi: 10.1029/2006JF000664.

C. Schoof. Ice-sheet acceleration driven by melt supply variability. *Nature*, 468(7325):803–806, 2010. doi: 10.1038/nature09618.

O. Sergienko and D. J. Wingham. Diverse behaviors of marine ice sheets in response to temporal variability of the atmospheric and basal conditions. *Journal of Glaciology*, pages 1–12, 2024.

O. V. Sergienko and D. J. Wingham. Bed topography and marine ice-sheet stability. *Journal of Glaciology*, page 1–15, 2021. doi: 10.1017/jog.2021.79.

A. Sommers, H. Rajaram, and M. Morlighem. Shakti: Subglacial hydrology and kinetic, transient interactions v1.0. *Geoscientific Model Development*, 11(7):2955–2974, 2018. doi: 10.5194/gmd-11-2955-2018.

V. C. Tsai, A. L. Stewart, and A. F. Thompson. Marine ice-sheet profiles and stability under Coulomb basal conditions. *Journal of Glaciology*, 61(226):205–215, 2015.

S. Tulaczyk, W. B. Kamb, and H. F. Engelhardt. Basal mechanics of Ice Stream B, West Antarctica I. Till mechanics. *Journal of Geophysical Research*, 105:463–482, 2000. doi: 10.1029/1999JB900329.

C. J. Van der Veen. The west antarctic ice sheet: the need to understand its dynamics. In C. J. Van der Veen and J. Oerlemans, editors, *Dynamics of the West Antarctic Ice Sheet*, pages 1–16. D. Reidel Publishing Company, 1987. Proceedings of a Workshop held in Utrecht, May 6-8, 1985.

J. S. Walder and A. Fowler. Channelized subglacial drainage over a deformable bed. *Journal of Glaciology*, 40(134):3–15, 1994. doi: 10.3189/S0022143000003750.

J. Weertman. On the sliding of glaciers. *Journal of Glaciology*, 3:33–38, Mar. 1957.

J. Weertman. Stability of the junction of an ice sheet and an ice shelf. *Journal of Glaciology*, 13:3–11, 1974.

L. K. Zoet and N. R. Iverson. A slip law for glaciers on deformable beds. *Science*, 368(6486):76–78, 2020. doi: 10.1126/science.aaz1183.